# Aggressive natural killer-cell leukemia mutational landscape and drug profiling highlight JAK-STAT signaling as therapeutic target

Olli Dufva[1], Matti Kankainen[2,3], Tiina Kelkka [1], Nodoka Sekiguchi [4], Shady Adnan Awad[1], Samuli Eldfors [2], Bhagwan Yadav[1], Heikki Kuusanmäki[1,2], Disha Malani[2], Emma I Andersson [1], Paavo Pietarinen[1], Leena Saikko[5], Panu E. Kovanen[5], Teija Ojala[6], Dean A. Lee[7], Thomas P. Loughran Jr.[8], Hideyuki Nakazawa[9], Junji Suzumiya[10], Ritsuro Suzuki[10], Young Hyeh Ko [11], Won Seog Kim[12], Shih-Sung Chuang[13], Tero Aittokallio [2], Wing C. Chan[14], Koichi Ohshima[15], Fumihiro Ishida[16] & Satu Mustjoki [1,17]

Aggressive natural killer-cell (NK-cell) leukemia (ANKL) is an extremely aggressive malignancy with dismal prognosis and lack of targeted therapies. Here, we elucidate the molecular pathogenesis of ANKL using a combination of genomic and drug sensitivity profiling. We study 14 ANKL patients using whole-exome sequencing (WES) and identify mutations in *STAT3* (21%) and RAS-MAPK pathway genes (21%) as well as in *DDX3X* (29%) and epigenetic modifiers (50%). Additional alterations include JAK-STAT copy gains and tyrosine phosphatase mutations, which we show recurrent also in extranodal NK/T-cell lymphoma, nasal type (NKTCL) through integration of public genomic data. Drug sensitivity profiling further demonstrates the role of the JAK-STAT pathway in the pathogenesis of NK-cell malignancies, identifying NK cells to be highly sensitive to JAK and BCL2 inhibition compared to other hematopoietic cell lineages. Our results provide insight into ANKL genetics and a framework for application of targeted therapies in NK-cell malignancies.

[1] Hematology Research Unit Helsinki, University of Helsinki and Department of Hematology, Helsinki University Hospital Comprehensive Cancer Center, FIN-00290 Helsinki, Finland. [2] Institute for Molecular Medicine Finland (FIMM), University of Helsinki, FIN-00014 Helsinki, Finland. [3] Medical and Clinical Genetics, University of Helsinki and Helsinki University Hospital, FIN-00290 Helsinki, Finland. [4] Department of Comprehensive Cancer Therapy, Shinshu University School of Medicine, Matsumoto 390-8621, Japan. [5] Department of Pathology, HUSLAB and Haartman Institute, University of Helsinki and Helsinki University Hospital, FIN-00290 Helsinki, Finland. [6] Pharmacology, Faculty of Medicine, University of Helsinki, FIN-00014 Helsinki, Finland. [7] Nationwide Children's Hospital, Division of Hematology, Oncology, and BMT, Columbus, OH 43205, USA. [8] Department of Medicine, University of Virginia, Charlottesville, VA 22908-0334, USA. [9] Division of Hematology, Internal Medicine, Shinshu University School of Medicine, Matsumoto 390-8621, Japan. [10] Department of Oncology/Hematology, Shimane University Hospital, Izumo 693-8501, Japan. [11] Department of Pathology, Samsung Medical Center, Seoul 0635, South Korea. [12] Sungkyunkwan University School of Medicine, Samsung Medical Center, Seoul 0635, South Korea. [13] Department of Pathology, Chi-Mei Medical Center, Tainan 71004, Taiwan. [14] Department of Pathology, City of Hope National Medical Center, Duarte, CA 91010, USA. [15] Department of Pathology, Kurume University School of Medicine, Kurume 830-0011, Japan. [16] Department of Biomedical Laboratory Sciences, Shinshu University School of Medicine, Matsumoto 390-8621, Japan. [17] Department of Clinical Chemistry, University of Helsinki, FIN-00014 Helsinki, Finland. These authors contributed equally: Matti Kankainen, Tiina Kelkka. Correspondence and requests for materials should be addressed to S.M. (email: satu.mustjoki@helsinki.fi)

Aggressive natural killer-cell (NK-cell) leukemia (ANKL) is a rare mature NK-cell neoplasm manifesting as a rapidly progressing systemic disease with an extremely poor median survival of just a few months[1,2]. The disease is highly resistant to treatment and currently available therapy options include chemotherapy and hematopoietic stem cell transplantation[3,4]. ANKL is most prevalent in the Asian population and known to be strongly associated with the Epstein-Barr virus (EBV) infection[4]. Little is known about the other aspects of the molecular pathogenesis of the disease, although some copy-number aberration analyses[5] and targeted sequencing of limited patient cohorts[6–8] have been performed. In contrast to ANKL, the closely related extranodal NK/T-cell lymphoma, nasal type (NKTCL), an extranodal lymphoma commonly presenting in the nasal cavity, has been more thoroughly studied. NKTCL can be distinguished from normal NK cells by deregulation of janus kinase-signal transducer and activator of transcription (JAK-STAT), AKT, and NF-κB signaling[9]. Recurrent chromosomal aberrations in NKTCL include a 6q21 deletion leading to the silencing of tumor suppressors PRDM1 and FOXO3[10]. Recently, mutations in the RNA helicase gene DDX3X were identified in 20%[11], and JAK-STAT pathway mutations, including STAT3 and STAT5B mutations[12–15], in a sizeable fraction of NKTCL patients. However, the exome-wide mutational landscape of ANKL has not been characterized.

Here, we investigate the mutational landscape of ANKL using WES and integrate these data to WES data from NKTCL and other related cancers to understand relationships between these diseases. Moreover, we characterize cell lines derived from NK cell neoplasms and normal NK cells using RNA sequencing and high-throughput drug sensitivity profiling to identify therapeutically actionable drivers in malignant NK cells. We report mutations in STAT3, the RAS-mitogen-activated protein kinase (RAS-MAPK) pathway, DDX3X and epigenetic modifiers in ANKL patients and demonstrate the importance of the JAK-STAT pathway in NK cells using drug sensitivity profiling, revealing potential therapeutic targets in NK-cell malignancies.

## Results

**Spectrum of somatic mutations in ANKL.** We performed WES on four tumor-normal pairs and ten tumor-only samples of ANKL to elucidate the molecular pathogenesis of ANKL (Supplementary Fig. 1, Supplementary Table 1, Supplementary Data 1, 2). To enable comparison to related cancers, we also reanalyzed published NKTCL WES data[11] and in-house WES data from three chronic lymphoproliferative disease of NK cells (CPLD-NK), 15 T-cell large granular lymphocytic leukemia (T-LGLL) and four T-cell prolymphocytic leukemia (T-PLL) patients using identical methods (Supplementary Fig. 1, Supplementary Data 3). The spectrum of single-nucleotide mutations in ANKL showed a preference for C>T, C>A and A>G substitutions, consistent with other cancers (Fig. 1a). However, comparison of the trinucleotide mutation signatures revealed differences, notably a relative absence of signature 3, associated with failure of DNA double-strand break repair by homologous recombination[16,17] (Fig. 1b). ANKL cases also largely clustered separately from the other tumor types by the spectrum of mutational signatures (Supplementary Fig. 2a). We also observed a higher mutation load in ANKL and NKTCL than in CLPD-NK, T-LGLL and T-PLL, although reaching statistical significance only between NKTCL and other cancers (Fig. 1c, Supplementary Fig. 2b). In addition, we detected a markedly higher fraction of reads mapping to the EBV genome in all tumor samples compared to controls, confirming the presence of EBV in the studied ANKL and NKTCL cases (Fig. 1d, Supplementary Fig. 2c). However, we did not

observe connections between EBV status and mutational signatures (Supplementary Fig. 2). The differences in the mutational signatures suggest that the underlying mutational processes in ANKL are at least partially different than those in related tumor types such as NKTCL.

**JAK-STAT, RAS-MAPK and epigenetic modifier mutations in ANKL.** In total, we identified several copy-number alterations (Supplementary Fig. 3) and 419 nonsynonymous somatic mutation candidates in tumor-control pairs and 529 in tumor-only samples (with an estimated sensitivity of 0.72 and positive predictive value of 0.36; Supplementary Data 1) involving 852 genes and including 298 missense, 63 nonsense, and 37 splice-site mutations as well as 131 frameshift and 81 inframe insertions or deletions. The complete lists of identified mutations are presented in Supplementary Data 2. Among the most recurrently mutated genes in ANKL were DDX3X (29%, 4/14 patients) and STAT3 (21%, 3/14 patients) (Fig. 1d). Markedly, 2/3 discovered STAT3 mutations have previously been reported as activating[12,18], and they localized to exons 20 and 21 encoding the Src homology 2 (SH2) domain mediating the dimerization and activation of STAT3 (Fig. 2a). Additionally, in one of the patients with a STAT3 mutation, we detected a 9p copy number gain containing JAK2 (Fig. 2b, Supplementary Fig. 3) and a point mutation in the protein tyrosine phosphatase (PTP) PTPRK, a commonly deleted tumor suppressor shown to negatively regulate STAT3 in NKTCL[19]. Other mutated PTPs included PTPN4 and PTPN23 (HD-PTP).

In addition to JAK-STAT aberrations, gain-of-function mutations in the RAS-MAPK pathway genes occurred in 3/14 ANKL patients, including those leading to the constitutive RAS activation (p.G12D and p.G13D) as well as a BRAF mutation (p.G469A). We also detected mutations in epigenetic regulators and histone-modifying enzymes in 7/14 patients, including mutations in BCOR and KMT2D (MLL2), mutated also in NKTCL[11,20], as well as in SETD2, mutated also in enteropathy-associated T-cell lymphoma[21,22]. One patient harbored a nonsense mutation in TET2 (p.Y899X), frequently inactivated in myeloid cancers and T-cell lymphomas[23]. Mutations in the RNA helicase gene DDX3X, commonly mutated in NKTCL, were found in four patients. These included a frameshift (p.T231fs), an inframe deletion, and two point mutations (p.L443V and p.A483G) localizing to the C-terminal helicase domain and predicted damaging (Supplementary Data 2). Other mutations in NKTCL-related genes included a nonsense FAS mutation, leading to truncation of the death domain[24] and a nonsense mutation in INPP5D[25]. One patient harbored a TP53 splice site mutation co-occurring with a 17p loss and a frameshift mutation in the DNA mismatch repair gene MSH6. Interestingly, this case had the highest mutation load among the tumor-normal ANKL cases and was the only ANKL case with evidence of mutation signature 3 (Supplementary Fig. 2a, b). Out of the identified genes with likely functional relevance, DDX3X, BCOR, and STAT3 were identified in ANKL as recurrent by both the MutSigCV and Oncodrive-fm algorithms (Fig. 1d, Supplementary Data 4). Moreover, these genes were abundantly expressed in NK cells, further supporting their role in ANKL pathogenesis (Fig. 1d, Supplementary Data 5).

**JAK-STAT and PD-L1 gains and PTP mutations in ANKL and NKTCL.** Given the discovery of JAK-STAT alterations in ANKL, we turned to analyze the extent of such alterations in NKTCL. By reanalyzing the NKTCL data[11], we discovered two previously undetected STAT3 mutations and a JAK2 mutation (p.V617F) in the NKTCL cohort, further expanding the fraction of NKTCL

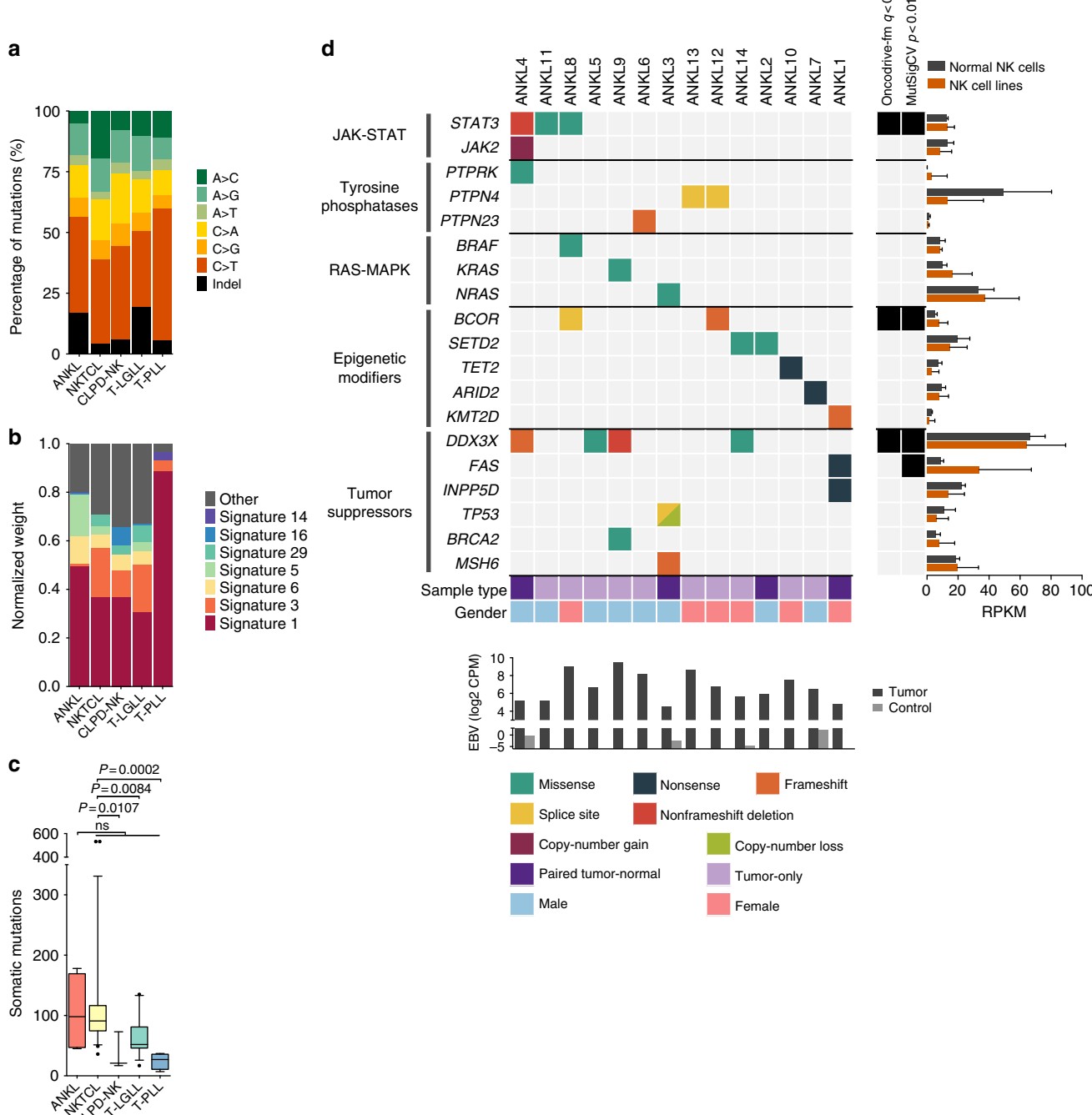

**Fig. 1** Whole-exome sequencing in ANKL. **a** Percentages of somatic base substitutions and indel mutations identified by whole-exome sequencing in tumor-normal paired samples of ANKL ($n = 4$), NKTCL ($n = 25$), CLPD-NK ($n = 3$), T-LGLL ($n = 15$) and T-PLL ($n = 4$). Synonymous mutations were included in the analysis. **b** Normalized weights of trinucleotide signatures identified using deconstructSigs in tumor-normal paired samples of ANKL, NKTCL, CLPD-NK, T-LGLL, and T-PLL. Weights of three most frequent signatures in each cancer type are shown across cancers as separate signatures and others are included under "other". Synonymous mutations were included in the analysis. **c** Numbers of somatic mutations in tumor-normal paired samples of ANKL, NKTCL, CLPD-NK, T-LGLL, and T-PLL. Synonymous mutations were included in the analysis. Horizontal lines indicate median, error bars indicate 10th and 90th percentiles, boxes represent interquartile ranges, and dots indicate outliers. *P* values were calculated using the Mann–Whitney *U*-test. **d** Alterations identified by whole-exome sequencing selected based on recurrence and biological significance. Complete lists of identified mutations are found in Supplementary Data 2. Diagonally split dual-colored rectangle indicates the presence of two alterations of different type in the same sample. Reads mapping to the EBV genome are reported as counts per million (CPM) under the figure. Results of the MutSigCV and Oncodrive-fm driver gene analyses are presented on the right side of the figure. Expression estimates of mutated genes in normal NK cells and NK cell lines are shown on the right as reads per kilobase per million mapped reads (RPKM), with bar length indicating mean and error bars representing range

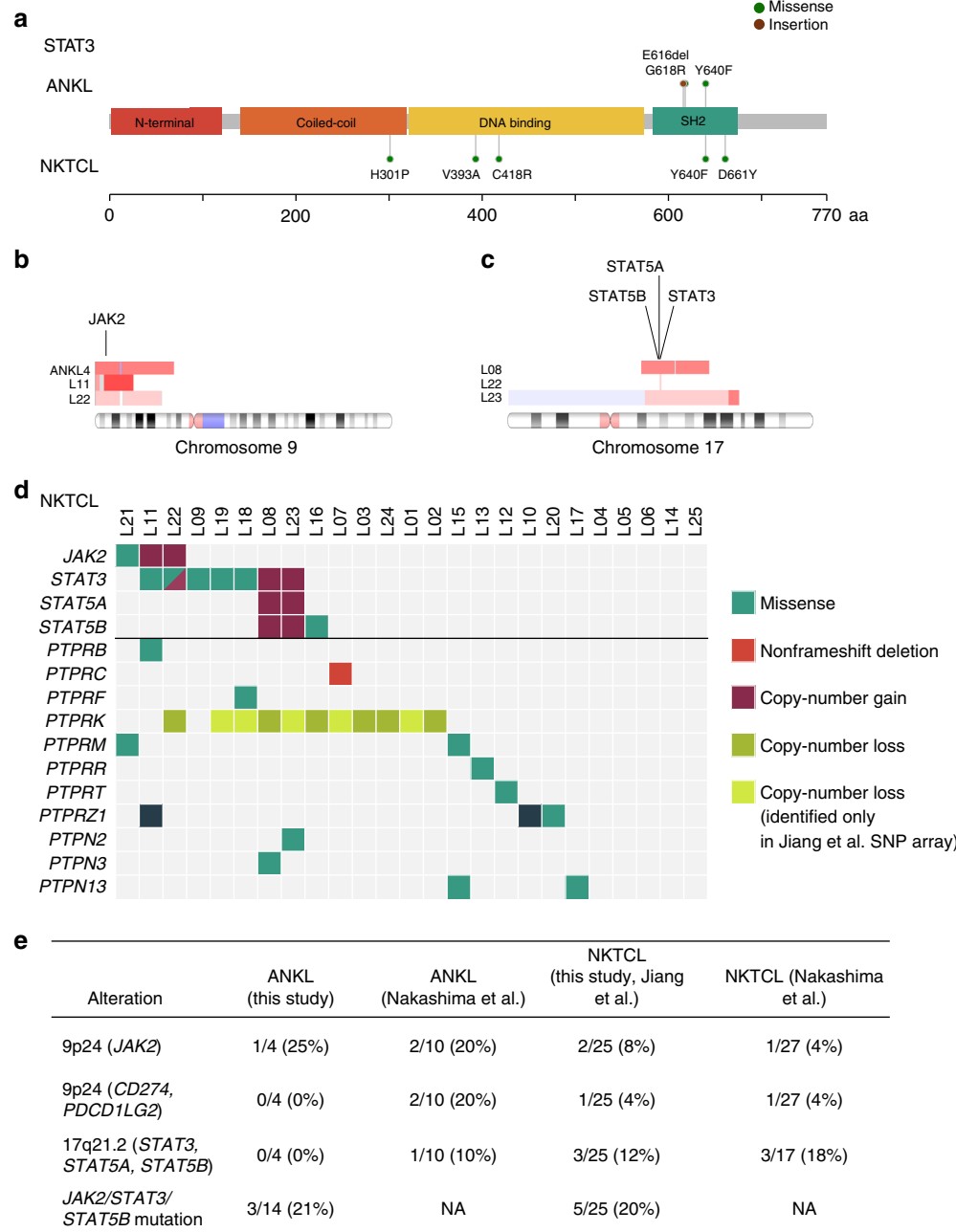

**Fig. 2** JAK-STAT pathway component alterations are common characteristics in both ANKL and NKTCL. **a** Somatic mutations identified in *STAT3* in ANKL (*n* = 14) and NKTCL (*n* = 25) patients. **b**–**c** Ideograms of chromosome 9 (**b**) and 17 (**c**) showing the areas of copy number alterations identified in ANKL and NKTCL. Red bars indicate gains and blue bars indicate losses. Patient IDs are indicated on the left and the locations of JAK-STAT genes are displayed. Only patients with gains are shown. **d** JAK-STAT and PTP alterations in NKTCL. Both alterations detected in this study using reanalyzed WES data and alterations detected by Jiang et al.[11] using SNP arrays are shown. Diagonally split dual-colored rectangle indicates the presence of two alterations of different type in the same sample. **e** Summary of JAK-STAT alterations identified in this study and in Nakashima et al.[5]

patients with JAK-STAT mutations (Fig. 2a). Furthermore, the reanalysis revealed various alterations other than point mutations affecting JAK-STAT signaling. Specifically, the *JAK2* region was amplified in 2/25 NKTCL cases (Fig. 2b, Supplementary Fig. 3b), with the copy gain spanning also *CD274* (PD-L1) and *PDCD1LG2* (PD-L2) in one case. Furthermore, we noted chromosome 17 gains in three NKTCL patients (12%), including one focal copy gain of *STAT3* and larger gains including also *STAT5A* and *STAT5B* in two cases (Fig. 2c, Supplementary Fig. 3b). In total, JAK-STAT amplifications were discovered in 16% of NKTCL cases (Fig. 2d). Review of the literature suggested recurrence of

these findings, uncovering 9p24 gains containing *JAK2*, *CD274* and *PDCD1LG2* in 2/10 ANKL and 1/27 NKTCL patients, as well as 17q21.2 gains containing *STAT3*, *STAT5A*, and *STAT5B* in 1/ 10 ANKL and 3/17 NKTCL cases[5] (Fig. 2e). Moreover, we found 8 receptor type and 3 non-receptor type PTPs to harbor mutations in 13/25 NKTCL cases in addition to the common 6q21 deletion containing *PTPRK* (Fig. 2d, Supplementary Table 2). Several of these genes have been implicated in negative regulation of JAK-STAT signaling, including *PTPRC* (CD45)[26], *PTPRT*[27], and *PTPN2* (TC-PTP)[28]. Most identified mutations were predicted damaging (Supplementary Table 2), suggesting a potential

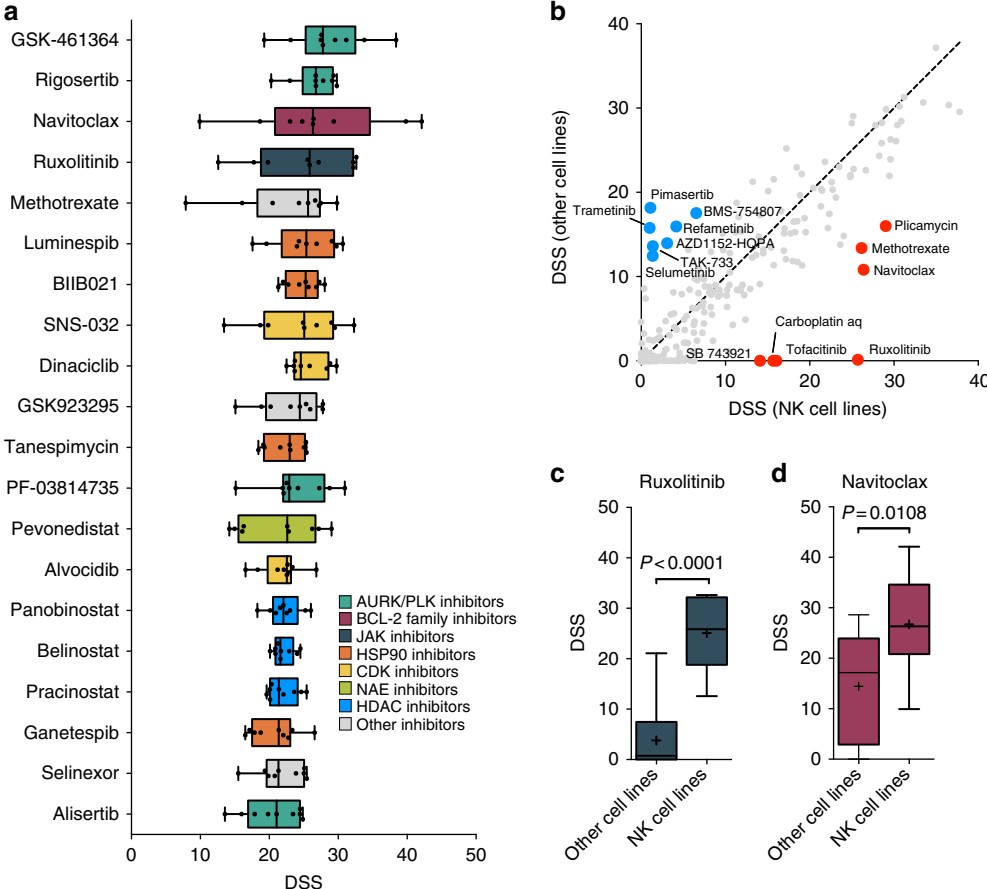

**Fig. 3** Effective targeted agents identified by drug sensitivity profiling in NK cell lines. **a** 25 most effective targeted compounds across NK cell lines ($n = 9$). Compounds are ordered by median drug sensitivity score (DSS). Higher DSS indicates higher sensitivity to a compound. Drugs annotated as conventional chemotherapeutics (Supplementary Data 7) and broadly active cytotoxic compounds (CUDC-907, YM155, daporinad, KX2-391, UCN-01, ONX-0914) are excluded. **b** Scatter plot comparing median DSS of NK cell lines ($n = 9$) to median DSS of other cell lines ($n = 29$), including 23 acute myeloid leukemia, three chronic myeloid leukemia, and three T-cell acute lymphoblastic leukemia cell lines, for 261 drugs. Drugs with median DSS > 10 units higher and lower in NK cells compared to other cells are labeled red and blue, respectively. **c**–**d** Comparison of DSS to ruxolitinib (**c**) and navitoclax (**d**) between NK cell lines and other cell lines. Lines in boxes indicate median, plus signs indicate mean, error bars indicate range and boxes represent interquartile ranges. $P$ values were calculated using the Mann–Whitney $U$-test

tumor suppressor role for PTPs in NK cells. The JAK-STAT copy gains and phosphatase mutations highlight previously unappreciated mechanisms of JAK-STAT alteration in NK-cell neoplasms.

**Drug sensitivity profiling in malignant NK cells**. Next, we aimed to understand how the commonly altered pathways in NK cells, importantly JAK-STAT, are reflected in drug response phenotypes. We subjected nine NK cell lines, including three ANKL and two NKTCL lines[29–37], to drug sensitivity profiling (Supplementary Table 3). Reassuringly, the mutational spectrum of these cell lines identified by RNA sequencing and targeted next-generation sequencing mimicked that of primary ANKL and NKTCL (Supplementary Fig. 4a, Supplementary Data 6). Mutations in STAT genes were detected in four cell lines, including a *STAT5B* mutation occurring at codon 642 (p.N642H) in DERL-7. In addition, mutations in epigenetic modifiers (*IDH1*, *BCOR*) targeting pathways mutated in ANKL[38] and tumor suppressors implicated in ANKL and NKTCL (*FAS*[24], *CASP8* belonging to the same apoptotic pathway, *DDX3X*, *MGA*[11] and *PRDM1*[10]) were discovered. Furthermore, transcriptomic comparison of the cell lines and primary NKTCL and T-LGLL as well as normal NK and T cells demonstrated the similarity of the cell lines to primary

NKTCL and normal NK cells rather than T-LGLL and normal T cells, suggesting that the transcriptomes of the cell lines reflect characteristics of the NK lineage and NK-cell malignancies (Supplementary Fig. 4b, c). Taken together, these results indicate utility of the cell lines as a useful model for drug sensitivity and resistance testing (DSRT).

In total, we quantified drug sensitivities of 459 approved and investigational oncology compounds over a 10,000-fold concentration range (Supplementary Data 7). Focusing on targeted agents, we found the JAK inhibitor ruxolitinib and the BCL2 family inhibitor navitoclax to be highly effective across the cell lines (Fig. 3a). Other effective drug classes were heat shock protein 90 (HSP90) inhibitors, Polo-like kinase (PLK) and Aurora kinase (AURK) inhibitors, cyclin-dependent kinase inhibitors as well as histone deacetylase inhibitors (Fig. 3a). Comparison of the drug sensitivity profile of NK cell lines to other hematologic cell lines, including acute and chronic myeloid leukemia and T-cell acute lymphoblastic leukemia cells, revealed that the JAK inhibitors ruxolitinib and tofacitinib were most selective towards NK cells, together with navitoclax and methotrexate also exhibiting NK-cell selectivity (Fig. 3b-d). In contrast, most NK cell lines were resistant to MEK inhibitors. Thus, the most highly active targeted agents in NK malignancies also show specificity towards NK cells compared

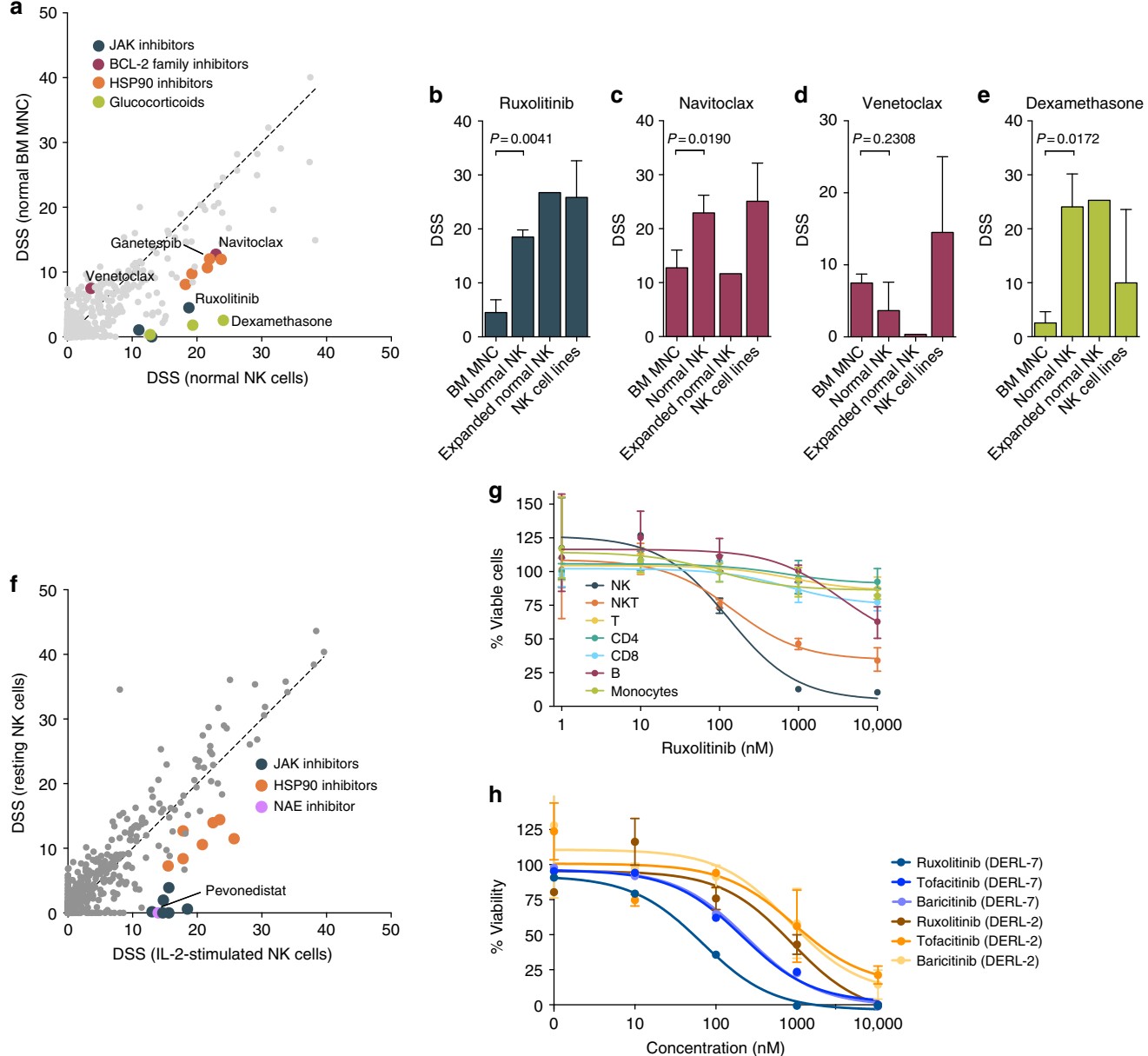

**Fig. 4** Drug response pattern characteristic of NK cells. **a** Drugs specific to normal NK cells compared to other hematopoietic cells. Scatter plot comparing median DSS of normal primary NK cells stimulated with 2.5 ng/mL IL-2 ($n = 3$) to median DSS of unstimulated normal bone marrow mononuclear cells (BM MNC, $n = 3$). **b**–**e** Comparison of drug responses of NK cells (normal IL-2-stimulated NKs, normal K562-aAPC-activated NKs and NK cell lines) and normal BM MNC to ruxolitinib (**b**), navitoclax (**c**), venetoclax (**d**) and dexamethasone (**e**). Bar height indicates mean DSS and error bars indicate standard deviation. $P$ values were calculated using Welch's $t$-test. **f** Inhibition of IL-2 signaling by JAK, HSP90 and NAE inhibitors. Scatter plot comparing DSS of resting ($n = 1$) and IL-2-stimulated normal primary NK cells ($n = 1$). **g** Ruxolitinib dose–response curves of different PBMC populations in the presence of 2.5 ng/mL IL-2. Shown are representative results from one out of two experiments. Dots indicate mean and error bars the range of three replicate wells. **h** JAK inhibitor dose–response curves of the cell lines DERL-7 (NK-cell characteristics) and DERL-2 (T-cell characteristics) established in parallel from the same patient with γδ lymphoma. Both cell lines were cultured in 2.5 ng/mL IL-2. Dots indicate mean and error bars the range of two independent experiments

to other malignancies, suggesting these drugs to target key biological pathways in NK cells.

**Specific sensitivity of NK cells to JAK and BCL2 inhibition.** To investigate in more detail whether the observed drug sensitivity pattern depended on the NK-cell phenotype and was similar to normal NK cells, we compared the drug responses of NK cells from healthy donors cultured in IL-2 to healthy bone marrow mononuclear cells (BM MNC) and NK cell lines. Healthy NK cells were more sensitive to JAK inhibitors and navitoclax as well

as HSP90 inhibitors than BM MNC, similar to what we observed in the cell lines (Fig. 4a-c). Responses to JAK inhibitors and navitoclax were similar in NK cell lines and normal NK cells that were cultured in IL-2 or additionally expanded using genetically engineered K562 cells[39] (Fig. 4b, c, Supplementary Fig. 5a, c). In contrast, the BCL2 inhibitor venetoclax and mTOR inhibitors were effective only in malignant NK cell lines, whereas responses to conventional chemotherapeutics were detected in both NK cell lines and expanded normal NK cells, likely reflecting their efficacy in actively proliferating cells (Fig. 4d, Supplementary Fig. 5).

Interestingly, glucocorticoids were highly effective in healthy NK cells compared to other cell types but induced responses only in few cell lines, implying glucocorticoid resistance in a subset of malignant NK cells (Fig. 4a, e, Supplementary Fig. 5a, c). Pairwise correlation of drug sensitivities revealed sensitivity to MEK inhibitors in the glucocorticoid-sensitive cells (Supplementary

Fig. 6). Taken together, we found JAK inhibitors and navitoclax uniformly effective in the NK lineage, whereas mTOR inhibitors and glucocorticoids induced responses only in cell lines or normal NK cells, respectively.

JAK and HSP90 inhibitors as well as the NAE inhibitor pevonedistat were more effective in IL-2-stimulated than resting

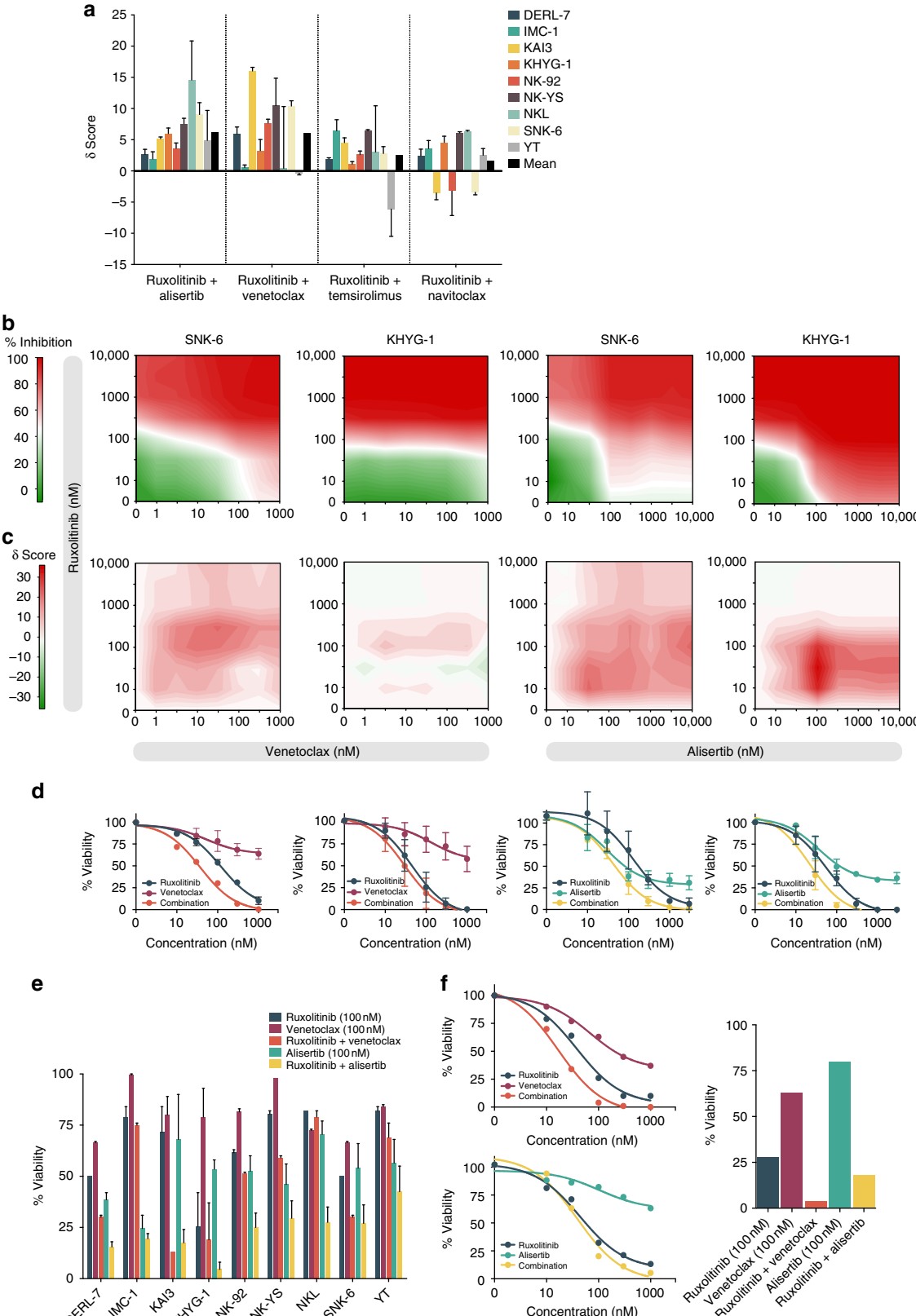

NK cells, suggesting that the sensitivity of NK cells to JAK and HSP90 inhibition resulted largely from inhibition of IL-2-derived JAK-STAT activation (Fig. 4f). This finding is consistent with the role of HSP90 in JAK-STAT regulation[40]. We therefore tested whether also other peripheral blood mononuclear cell (PBMC) populations acquired sensitivity to JAK inhibition upon IL-2 stimulation. NK cells, and to a lesser extent NKT cells, were markedly more sensitive to ruxolitinib than other hematopoietic cell types in the presence of IL-2, including T cells for which IL-2 is a key growth factor[41] (Fig. 4g). Moreover, the γδ lymphoma line DERL-7 harboring NK-cell characteristics was more sensitive to JAK inhibitors compared to the sister T-cell line DERL-2 established in parallel from the same patient[29] (ruxolitinib IC50 45.4 nM vs. 396.8 nM, respectively) (Fig. 4h). We conclude that NK cells show particular sensitivity to inhibition of IL-2-JAK-STAT signaling compared to other hematopoietic cells.

Given the effectiveness of JAK inhibition and strong genetic evidence for JAK-STAT pathway involvement, we tested whether combining ruxolitinib with other promising compounds found selective in normal NK cells or NK cell lines would lead to enhanced in vitro activity. We tested ruxolitinib in combination with the BCL2 family inhibitors navitoclax and the clinically approved venetoclax as well as the mTOR inhibitor temsirolimus and the AURK inhibitor alisertib, in dose–response matrices to identify synergistic relationships. Comparison of delta scores, a measure of the observed combination effect beyond the expected interaction between drugs[42], revealed synergistic behavior of venetoclax and alisertib across the cell lines when combined with ruxolitinib (Fig. 5a). Variation in the degree of synergy was observed between cell lines and drug combinations, with the JAK/AURK inhibitor combination being more effective in some cell lines (e.g. NKL) and the JAK/BCL2 inhibitor combination in others (e.g. KAI3) and some exhibiting equal efficacy (e.g. SNK-6) (Fig. 5a–d, Supplementary Fig. 7, 8). Interestingly, the BCL2-selective venetoclax was less effective as a single agent compared to the more broad-spectrum BCL2 family inhibitor navitoclax (Fig. 4c, d), but it increased the sensitivity to ruxolitinib more consistently. Importantly, venetoclax and alisertib potentiated the efficacy of ruxolitinib at clinically achievable nanomolar concentrations in the cell lines (Fig. 5e). In primary NK cells from a patient with bone marrow NK lymphoproliferation harboring an activating STAT3 mutation at codon 661 (p.D661Y), venetoclax showed higher efficacy in combination with ruxolitinib compared to alisertib (Fig. 5f). These results suggest that combination of other targeted agents to JAK inhibition may increase efficacy compared to single-agent treatment.

## Discussion

In this study, we used integrated genomic and drug sensitivity profiling to study the somatic mutation landscape of ANKL and identify therapeutic targets in NK-cell malignancies. We identified alterations in JAK-STAT, RAS-MAPK and epigenetic modifier genes as well as in DDX3X in ANKL and discovered high sensitivity of the NK lineage to JAK and BCL2 inhibition.

We found JAK-STAT signaling components to be frequently altered in ANKL, with 3/14 patients harboring a STAT3 mutation. Previous reports have described STAT3 mutations in two cases of EBV-negative ANKL[8] and a patient with a STAT5B mutation[6]. Gain-of-function STAT3 mutations have also been previously described in diseases of varying aggressiveness, such as NKTCL and the relatively indolent T-LGLL and CLPD-NK[18,43]. This suggests that deregulated JAK-STAT signaling alone is unlikely to explain the aggressive course of ANKL, but that other events such as alterations in epigenetic modifiers or interplay with EBV-associated factors are likely involved. Using published NKTCL and ANKL data, we found JAK2, STAT3, and STAT5B gains as well as previously undetected JAK2 and STAT3 mutations in NKTCL, extending the proportion of cases with JAK-STAT alterations. Together, our findings in ANKL and NKTCL strengthen the role of JAK-STAT alterations in NK-cell malignancies.

Furthermore, the JAK2 gain was associated with gains of the neighboring immune evasion-associated CD274 and PDCD1LG2 genes in a minority of both ANKL and NKTCL patients. Together with the PD-L1 induction associated with EBV infection[44], this suggests potential for PD-1 blockade immunotherapy also in a subset of ANKL patients in addition to recently reported sensitivity of NKTCL to PD-1 inhibition[45].

Uncovering the genetic landscape of ANKL enables assessment of the relationships of pathogenetic mechanisms between related cancers at the molecular level. In particular, clinicopathologic similarities between ANKL and NKTCL have raised questions that ANKL might represent the leukemic phase of NKTCL. Genes reported as frequently mutated in NKTCL, including DDX3X[11], STAT3[11,12], BCOR[20], KMT2D[11], and FAS[24], harbored mutations in our ANKL cohort, implying that similar pathogenetic mechanisms are present in both diseases. Jiang and colleagues found the co-occurrence of DDX3X and TP53 mutations to confer a worse prognosis and correlate with more aggressive phenotype in NKTCL[11]. However, these mutations were not overrepresented in ANKL compared to NKTCL, with TP53 mutations found in only one ANKL case, arguing against the hypothesis that ANKL would represent a more advanced form of NKTCL. Moreover, our ANKL cohort was relatively more enriched in RAS-MAPK pathway alterations than previous NKTCL cohorts, although the sample sizes are limited for definitive conclusions. Excluding these borderline differences, we did not discover specific mutations that would aid in differential diagnosis between ANKL and NKTCL. Our analysis of mutational signatures, however, revealed the lack of a DNA double-strand break repair-associated signature in ANKL compared to NKTCL, CLPD-NK, and T-LGLL. This implies that although several driver mutations appear similar between ANKL and NKTCL, the underlying mutational processes may differ. We did not identify mutational signatures correlating with the presence of EBV, and mutations recurrent among our EBV-positive cases, such as STAT3 and DDX3X, have also been reported to occur in EBV-negative cases[7,8], suggesting that EBV positivity does not

**Fig. 5** Drug combination strategies in NK cells. **a** Delta scores representing the degree of synergy (see Methods) of the selected drug combinations in NK cell lines. Bar heights indicate mean, and error bars the range of two independent experiments. Drug combinations are ranked according to mean delta score. **b** Dose–response matrices of percent inhibition achieved at indicated doses of ruxolitinib combined with venetoclax or alisertib in cell lines SNK-6 and KHYG-1. Shown are representative results from one out of two experiments. **c** Dose–response matrices of delta synergy scores achieved at indicated doses of ruxolitinib combined with venetoclax or alisertib in cell lines SNK-6 and KHYG-1. Shown are representative results from one out of two experiments. **d** Dose–response curves of percent inhibition achieved at indicated doses of ruxolitinib combined with venetoclax or alisertib in cell lines SNK-6 and KHYG-1. Dots indicate mean and error bars the range of two independent experiments. **e** Viability percentages for ruxolitinib, venetoclax and alisertib as single agents and in combination across NK cell lines normalized to DMSO and BzCl controls. **f** Dose–response curves and viability percentages for ruxolitinib, venetoclax, alisertib and combinations in primary NK cells isolated from a patient with a BM NK-cell lymphoproliferation harboring a STAT3 mutation

strikingly influence the mutational profile. The epigenetic modifier mutations discovered in ANKL suggest similar pathogenetic mechanisms in a subset of ANKL as in related mature T-cell neoplasms harboring similar alterations, such as in *SETD2* and *TET2*[21,22,46]. Furthermore, the transcriptional profile of NKTCL has been shown to resemble a subset of hepatosplenic γδ lymphoma, further highlighting the similarity of diseases within these diagnostic groups[47]. Taken together, the genetic etiology of ANKL appears rather heterogeneous and partially overlapping with several related diseases, and further studies are required to better understand the contribution of genetic alterations to clinical presentation.

Drug sensitivity profiling highlighted JAK inhibitors as effective targeted therapy candidates, in line with the central role of IL-2 signaling in NK cells and the JAK-STAT pathway alterations observed in ANKL and NKTCL. The JAK inhibitor ruxolitinib is approved for the treatment of myeloproliferative neoplasms and has been shown to impair NK cell function in vivo[48]. Moreover, we identified the BCL2 family inhibitor navitoclax as an effective compound across the NK cell lines and being selective towards NK cells compared to other hematopoietic cells. Navitoclax was previously shown to enhance JAK inhibition in another IL-2-dependent malignancy, adult T cell leukemia[49]. Although the BCL2 inhibitor venetoclax did not show comparable efficacy against NK cells in vitro as a single agent, it was able to potentiate response to ruxolitinib more consistently than navitoclax. Navitoclax treatment results in dose-dependent thrombocytopenia due to BCL-$X_L$ inhibition as well as T-cell lymphopenia, the latter of which is potentially related to the efficacy in the closely related NK cells found in this study[50]. In contrast, venetoclax, approved for treatment of chronic lymphocytic leukemia, avoids thrombocytopenia by higher BCL2 specificity[51]. Combining JAK and BCL2 inhibition in NK-cell malignancies could present a promising treatment strategy exploiting two NK-cell selective approaches, with venetoclax less likely to result in severe toxicities. Furthermore, AURK inhibitors showed efficacy as both single agents and in combination with ruxolitinib. This is consistent with previously identified sensitivity of NKTCL to an AURK inhibitor that induced responses also in our data, although rather modest compared to more potent AURK inhibitors[47] (Supplementary Data 7). However, the AURK inhibitor alisertib had limited efficacy as a single agent and in potentiating sensitivity to ruxolitinib against primary cells from a patient with STAT3-mutated NK-cell lymphoproliferation as well as in normal NK cells, suggesting that alisertib may be particularly effective in highly proliferating cells such as cell lines and expanded primary NK cells. In contrast to JAK, BCL2, and AURK inhibition, NK cells were relatively resistant to MEK inhibitors, which were effective only in a subset of samples. This suggests that sensitivity to MEK inhibition occurs only in selected cases and is not a unifying property of malignant NK cells. Our functional in vitro drug profiling approach thus demonstrates the utility of discerning essential druggable targets among various activated signaling processes.

In conclusion, this study provides understanding of the landscape of somatic mutations in ANKL and its relation to other similar cancers. Together with the integrated drug sensitivity profiling, our results highlight the central role of the JAK-STAT pathway and indicate novel possibilities for targeted therapies in NK-cell malignancies with notoriously poor prognosis.

## Methods

**Patients.** The clinicopathologic characteristics of ANKL patients ($n = 14$) are summarized in Supplementary Table 1. Patients diagnosed with ANKL and with available samples of sufficient quality and quantity for next-generation sequencing were identified from Shinshu University, Japan, Shimane University, Japan,

Fukuoka University Hospital, Japan, Chi-Mei Medical Center, Taiwan and Samsung Medical Center, South Korea. ANKL diagnosis was made according to the WHO classification, with a surface immunophenotype characteristic of NK cells ($CD2^+CD3^-CD56^+$), systemic involvement including bone marrow and/or peripheral blood and an aggressive clinical course as minimum requirements for diagnosis. Cases ANKL6–10 have been included in previous studies[5,10] (Supplementary Table 1). Informed consent was obtained from all patients in accordance with the Declaration of Helsinki and the study was approved by the ethics committee of Shinshu University School of Medicine.

**Cell lines.** Characteristics of NK cell lines ($n = 9$) included in this study are summarized in Supplementary Table 3. DERL-2, DERL-7, KHYG-1, NK-92, and YT cell lines were obtained from the Deutsche Sammlung von Mikroorganismen und Zellkulturen GmbH (DSMZ). Cell lines KAI3, SNK-6, and NK-YS were obtained from Dr. Wing C. Chan (City of Hope Medical Center, CA) and NKL from Dr. Thomas P. Loughran (University of Virginia, VA). IMC-1 cell line was a generous gift from Dr. I-Ming Chen (University of New Mexico, NM). For cell lines obtained from cell bank (above) or for which authentication was available (DERL-2, DERL-7, KHYG-1, KAI3, and NK-92), experiments were performed within 20 passages of obtaining cells from the cell bank or authentication. Authentication was performed using GenePrint10 System (Promega). Cell lines were not tested for mycoplasma contamination. All cell lines were cultured in RPMI-1640 (Lonza) with 10% FBS, 2 mM L-glutamine, 100 U/mL penicillin, and 100 μg/mL streptomycin (R10). Culture medium was supplemented with 2.5 ng/mL recombinant human IL-2 (Peprotech) except in case of IMC-1, for which 20 ng/mL was used.

**Primary cell isolation for DSRT and RNA sequencing.** Primary human NK cells were isolated from peripheral blood mononuclear cells (PBMC) obtained by Ficoll-Paque separation from buffy coats provided by the Finnish Red Cross. NK cells were isolated using a human NK cell isolation kit (Miltenyi Biotec). NK cells from three donors were pooled to achieve enough material for DSRT and RNA sequencing. The purity of NK cells was evaluated by flow cytometry using CD56-PE (clone NCAM16.2) and CD3-APC (clone SK7) staining (BD Biosciences), and the final pool contained >90% $CD56^+CD3^-$ cells in all replicates. Primary bone marrow mononuclear cells (BM MNC) were isolated from bone marrow aspirates from healthy donors using Ficoll-Paque separation. All normal NK cells were cultured in R10 supplemented with 2.5 ng/mL recombinant human IL-2 (Peprotech). BM MNC were cultured in Mononuclear Cell Medium (PromoCell). To obtain actively proliferating normal NK cells, NK cells were expanded using an artificial antigen-presenting cell K562 variant (K562-aAPC) as previously described[39]. Briefly, PBMC were isolated from buffy coats by Ficoll-Paque separation and co-cultured with irradiated (100 cGy) K562-aAPCs at a ratio of 1:2 (PBMC:aAPC) in RPMI1640 with 50 IU/mL IL-2 at 200,000 PBMC/mL. Cultures were refreshed with half-volume media changes every two to three days, and re-stimulated with aAPCs at ratio of 1:1 every seven days.

**Whole-exome sequencing and targeted DNA sequencing.** For tumor-normal samples, genomic tumor DNA was extracted from either total PBMC/BM MNC when tumor content was over 90% or from $CD3^-CD56^+$ cells sorted using FACSAria (BD Biosciences) for samples with lower tumor content. Germline DNA was obtained either from healthy non-hematopoietic tissue or NK cell-negative fraction of sorted PBMC/BM MNC. (Supplementary Data 1). Exome capture was performed using Nextera Rapid Capture Exome Kit (Illumina). In the case on tumor-only samples, genomic tumor DNA from samples was extracted from total BM MNC or formalin-fixed paraffin-embedded bone marrow (sample ANKL11) without enrichment for tumor cells. Exome capture was then performed using Agilent SureSelect XT Clinical Research Exome kit (Agilent). Finally, genomic DNA was extracted from cell lines and targeted sequencing was performed using a target enrichment SeqCap EZ Comprehensive Cancer Design panel (Roche NimbleGen) comprising 578 cancer genes. In all cases, sequencing libraries were sequenced using paired end 100 bp read format on an Illumina HiSeq 2000 instrument (Illumina).

**Variant analysis.** SRA accession identifiers for public datasets are listed in Supplementary Data 1. Pre-processing of short read data was done using the Trimmomatic software[52] and included correction of the sequence data for adapter sequences, bases with low quality, and reads less than 36 bp in length. Paired-end reads passing the pre-processing were aligned to human reference genome build 38 (EnsEMBL v82) using BWA-MEM[53] with default parameters. Reads were then sorted by coordinate using the SortSAM and PCR duplicates were marked using the MarkDuplicate module of the Picard toolkit (Broad Institute). Calling of variants was done using the Genome Analysis Toolkit (GATK) toolset[54] and GATK resource files that were converted from GRCh37 to GRCh38 using CrossMap[55] and EnsEMBL chain files downloaded from EnsEMBL. Briefly, local indel realignment was performed around indels using GATK IndelRealigner and base qualities were recalibrated using GATK BaseRecalibrator[56], and the cross-sample contamination level was estimated using GATK4 CalculateContamination. Variants were then called using GATK Mutect2 by using the observed cross-sample contamination

level and filtered against a panel of normals generated from the exome data of 24 healthy unrelated Finnish individuals. Finally, the levels of 8-oxoguanine and deamination artifacts were estimated using GATK4 CollectSequencingArtifact-Metrics and GATK4 FilterByOrientationBias was used to remove these artifacts. Version information of tools used in the variant calling steps are given in Supplementary Data 1. Accuracy of the tumor-only variant calls by comparison with the matched tumor-control variant calls is given in Supplementary Data 1, revealing an average sensitivity of 0.72 and positive predictive value of 0.36. Sequencing coverage was evaluated by examining reads mapped to exons of protein coding genes (EnsEMBL v82) with 150 bp padding on each side of the exons using bedtools coverage software[57], revealing a mean coverage of 48x and 72x across the ANKL tumor and germline samples, respectively, with 63% of the target intervals with sequence coverage being covered by at least 10x (Supplementary Data 1).

Annotation and filtering of variants was done using the Annovar tool[58] against the RefGene database. Briefly, only variants passing all MuTect2 filters with a TLOD ≥ 6.3 alone or a TLOD ≥ 5.0 and supported by two or more independent COSMIC[59] samples were taken into account. For the trinucleotide profile and driver gene analyses, the initial set of somatic variants was filtered for false-positives by removing variants with a minor allele frequency ≥1% in the EPS, 1KG, general ExAC (ExAC), East Asian ExAC (ExAC_EAS), Finnish ExAC (ExAC_FIN) databases, coverage ≤10, and variant quality value ≤40. For other analyses, tumor-normal somatic variant calls were further filtered by removing synonymous mutations, while tumor-only samples were further filtered by accepting only mutations occurring at a frequency <0.01% in the ExAC and ExAC_EAS databases or mutations without ExAC information, but supported by ≥2 independent COSMIC samples and assumed to impair protein function (i.e. nonsense and indel mutations and mutations predicted as damaging by five or more of the nine variant effect predictors used; Supplementary Data 1). Finally, mutations included in Fig. 1d based on their biological significance were manually inspected using Integrative Genomics Viewer 2.3.66 (Broad Institute).

Identification of mutational signatures within tumor samples was done using the deconstructSigs[17] software with default parameters and cancer profiles downloaded from the COSMIC web site on September 2017. In the analysis, conversion of EnsEMBL to UCSC chromosome nomenclature was done using the function mapSeqlevels from the package GenomeInfoDb. Potential driver genes were identified using MutSigCV[60] and Oncodrive-fm[61]. MutSigCV was executed using the default mutation rate covariate after fixing its gene names to current nomenclature with maftools[62], a hg38-compatible coverage file generated with the CovGen tool, and the default mutation type dictionary file after addition of Annovar variant categories into it. For Oncodrive-fm, functional impact scores of missense mutations were gathered from the PolyPhen2 (HumVar), SIFT, and MutationAssessor annotations of Annovar outputs. Scores were manipulated in accordance with recommendation[61] and Oncodrive-fm executed with default setting except for setting the minimum number of mutations per gene to 3. A step-by-step documentation of the variant analysis and version information of tools used are available in Supplementary Data 1.

**Copy-number alteration analysis**. SRA accession identifiers for public datasets are listed in Supplementary Data 1. Raw reads were merged with SeqPrep. Resulting paired reads were trimmed of B blocks in the quality scores from the end of the read. Trimmed reads shorter than 36 base pairs were removed. Reads were aligned using the Burrows–Wheeler Aligner against the human genome GRCh37 reference-genome primary assembly, reads mapping to multiple genomic positions were removed, alignments were refined using GATK Indel Realignment[56], and potential PCR duplicates filtered by using Picard MarkDuplicates. All exome sequencing capture kit target regions that were less than 76 bp apart were then merged with each other. An FPKM (fragment per kilobase of target region length per million mapped reads) copy number value was calculated separately for each merged target region. We then filtered out the regions with sequencing coverage lower than 25×. Finally, the log2 copy number ratios for each sample were divided by the reference and calculated and segmented using Circular Binary Segmentation[63]. The copy number data for all human genes in the Ensembl database v67 were calculated by assigning a gene the value of the copy number variant data segment with which it overlapped. If a gene overlapped with more than one segment, the gene was given the lowest segment log2 value when an overlapped segment had a log2 ratio <−0.6 and the highest segment value when an overlapped segment had a log2 ratio >0.5. If all segments of the overlapping gene had a log2 ratio >−0.6 and <0.5, the gene was assigned the median log2 ratio of all the overlapping segments.

**Pathogen discovery**. All datasets were pre-processed as described above in variant analysis. Microbial classification of pre-processed paired-end reads was then done using Centrifuge[64]. The uncompressed p+h+v Centrifuge index (12/06/2016 version) was used in the process comprising 28718 viral, prokaryotic, and human genomes and technical artifact sequences. Read counts were converted to counts per million (CPMs) by dividing them by the total number of reads of the root in millions. The sensitivity and specificity of the method was established by analyzing NK cell line data, revealing the scarcity of EBV reads (CPM < 1) in EBV negative NK cell and presence of significant amounts of EBV reads (CPM > 64) in NK cells known to have an EBV infection (Supplementary Fig. 4).

**RNA sequencing**. Cell lines and healthy NK cells were seeded at $5 \times 10^5$ cells/mL and cultured for 72 h in R10 supplemented with 2.5 ng/mL IL-2 (20 ng/mL for IMC-1) prior to harvesting for extraction. RNA was extracted using miRNeasy Mini Kit (Qiagen). Agilent Bioanalyzer RNApico chip (Agilent) was used to evaluate the integrity of RNA and Qubit RNA kit (Life Technologies) to quantitate RNA in samples. 1.5 μg of total RNA was used for ScriptSeq™ Complete Kit for human/mouse/rat (Illumina) to ribodeplete rRNA and further for RNA-seq library preparation. SPRI beads (Agencourt AMPure XP, Beckman Coulter) were used for purification of RNAseq libraries. The library QC was evaluated on High Sensitivity chips by Agilent Bioanalyzer (Agilent). Sequencing libraries were sequenced using paired end 100 bp read format on an Illumina HiSeq 2000 instrument (Illumina).

RNA sequencing data were pre-processed similarly to DNA sequencing data. SRA accession identifiers for public datasets are listed in Supplementary Data 1. Paired-end reads passing the pre-processing were then aligned to human reference genome build 38 (EnsEMBL v82) using STAR[65] with the default 2-pass per-sample mapping settings. Reads were then sorted by coordinate using the SortSAM and PCR duplicates were marked using the MarkDuplicate module of the Picard toolkit. Calling of variants was done according to the GATK best practice for calling variants on RNA sequencing data, including splitting of pre-processed and mapped reads into exon segments using GATK SplitNCigarReads, local indel realignment around indels using GATK IndelRealigner, and base quality recalibration using GATK BaseRecalibrator. Variant calling relied on the GATK HaplotypeCaller and variants were filtered using GATK VariantFiltration according to the best practice recommendations regarding the RNA-seq variant analysis workflow. To reduce the number of false positive variants, variants were annotated and filtered as DNA sequencing variants, except that variants had to pass all default variant caller filters without relaxation in their TLOD scores. To identify potentially disease-causing RNA-seq variants, analyses were restricted to protein function-altering mutations (nonsense and frameshift variants) and variants supported by two or more independent COSMIC samples. In transcriptome comparisons, mapped reads were assigned to gene features (EnsEMBL v82) using FeatureCounts[66] by allowing multi-mapping reads and assignment of a read to more than one overlapping feature. Read counts were normalized by Trimmed Mean of $M$-values (TMM) normalization[67] and CPM and RPKM (reads per kilobase per million mapped reads) values were computed using edgeR[68] with default parameters. Data for the transcriptomic comparison of NK cell lines and normal NK cells and data for the transcriptomic comparison of the cell lines and primary NKTCL and T-LGLL as well as normal NK and T cells were normalized separately.

**Drug sensitivity and resistance testing (DSRT)**. The oncology compound collection included 145 FDA/EMA approved anti-cancer and other drugs and 314 investigational and preclinical compounds (Supplementary Data 7). All compounds were purchased from commercial chemical vendors and dissolved in either 100% dimethyl sulfoxide (DMSO) or water. DSRT was performed as previously described[69]. Briefly, each compound was preprinted on 384-well plates (Corning) in five different concentrations covering a 10,000-fold concentration range with an acoustic liquid handling device (Echo 550, Labcyte Inc.). Compounds were dissolved in 5 μl culture medium on a shaker for 10 min. 20 μl of single-cell suspension of cell lines (3000 cells per well) or primary cells (10,000 cells per well) were dispensed using Multi-Drop Combi peristaltic dispenser (Thermo Scientific) or MultiFlo FX Multi-Mode Dispenser (BioTek). Plates were incubated at 37 °C and 5% $CO_2$ for 72 h after which cell viability was measured using CellTiter-Glo 2.0 reagent (Promega) according to the manufacturer's instructions with a Pherastar FS plate reader (BMG Labtech). Cell viability luminescence data were normalized to DMSO-only wells (negative control) and 100 mM benzethonium chloride-containing wells (positive control). The DSRT data were quantified using the drug sensitivity score (DSS)[69,70]. For comparison to other cell lines screened in-house, a common drug collection of 261 overlapping compounds between all cell lines was used (Supplementary Data 7).

**Flow cytometry**. For profiling of ruxolitinib sensitivity of PBMC cell populations, PBMC from buffy coats were plated on V-bottom 96-well plates at 100,000 cells/well in 100 μl R10 supplemented with 2.5 ng/mL IL-2 in the presence of indicated concentrations of ruxolitinib or DMSO as a control, all conditions in triplicate. After 72 h incubation at 37 °C and 5% $CO_2$, cells were centrifuged, resuspended to 25 μl staining buffer (PBS + 0.5% BSA + 0.02% $NaN_3$) and stained with antibodies for CD56-FITC (clone NCAM16.2), CD4-PE-Cy7 (clone RPA-T4), CD3-APC (clone SK7), CD8-APC-H7 (clone SK1), CD14-V500 (clone M5E2) (all from BD Biosciences), and CD19-Pacific Blue (clone SJ25-C1, Invitrogen) for 15 min at RT. Cells were then washed with 100 μl staining buffer and resuspended to 25 μl Annexin V binding buffer with AnnexinV-PE and 7-AAD (both from BD Biosciences). Cells were acquired using an iQue Screener Plus flow cytometer (Intellicyt) and cell populations were gated as shown in Supplementary Figure 9 by first gating for viable cells based on AnnexinV/7-AAD negativity and then gating PBMC subpopulations. B cells were gated as CD19$^+$ and monocytes as CD14$^+$ cells. Counts of viable cells in each population per well were normalized to counts in DMSO-only wells (100% viability) and zero viable cells (0% viability).

**Drug combination analysis**. Cells were plated on 384-well plates with dose combination matrices comprising seven different concentrations for each drug with DMSO controls and DSRT was performed as described above. Drug combination effect was quantified by comparing the observed joint inhibition level at each dose combination to the expected combination effect using the zero-interaction potency (ZIP) model using previously published scripts[42]. The ZIP model assumes that two non-interacting drugs are expected to incur minimal changes in their drug response curve. It combines the advantages of Loewe additive model and Bliss independence model. Delta score is an average combination effect of the drugs over all the tested concentrations. The delta score 0 implies both probabilistic independence and dose additivity. The delta score >0 or <0 is synergistic or antagonistic effect of drugs, respectively.

**Statistical analysis**. Mann–Whitney U-test and Welch's t-test were applied using GraphPad Prism 6 software as indicated in the figure legends according to assumptions on data normality. False discovery rate (FDR) approach to correct for multiple comparisons was used where indicated using the corr.test function of the 'psych' package in R.

**Code availability**. Computer code used to generate results is available from authors upon suitable request.

**Data availability**. The RNA sequencing data from the cell lines have been deposited at the GEO database under the accession number GSE106391. All other patient whole-exome sequencing and RNA-seq data are available from the corresponding author upon suitable request.

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

## Acknowledgements

We thank the staff of High Throughput Biomedicine (Laura Turunen, Maria Nurmi, Jani Saarela) and Sequencing Laboratory (Pekka Ellonen, Sari Hannula, Aino Palva, Pirkko Mattila) Units at the Institute of Molecular Medicine Finland (FIMM) and Judith Klievink and other personnel of Hematology Research Unit Helsinki for their excellent technical assistance. We thank Dr. I-Ming Chen for kindly providing the IMC-1 cell line and Dr. Norio Shimizu for originally providing the SNK-6 cell line. This work was supported by the European Research Council (M-IMM project), Academy of Finland (grant no. 292605 and 287224), Tekes – the Finnish Funding Agency for Innovation (Dnro 6113/31/2016), Finnish special governmental subsidy for health sciences, research and training, the Sigrid Juselius Foundation, the Instrumentarium Science foundation, the Cancer Society of Finland, and the Finnish Cancer Institute. W.C.C. receives support from the City of Hope Cancer Center Support Grant (P30CA33572) and Lymphoma SPORE developmental project grant (1 P50 CA 136411-01 01A1 PP-4). F.I. was supported by KAKEN 15K09471 from Grant-in-Aid for Scientific Research from the Ministry of Education, Culture, Sports, Science and Technology of Japan; Japan Leukemia Research Fund; Takeda Pharmaceutical (inst); Cyugai Pharmaceutical (inst); Kyowa Hakko-Kirin (inst) and Pfizer (inst). The authors would like to thank the Exome Aggregation Consortium and the groups that provided exome variant data for comparison. A full list of contributing groups can be found at http://exac.broadinstitute.org/about.

## Author contributions

O.D. designed the study, performed experiments and analyzed WES, RNA sequencing and drug sensitivity data. M.K. and S.E. performed WES and RNA sequencing analyses. T.K. participated in the study design and performed experiments. S.A.A., H.K., D.M., E.I.A., and P.P. performed experiments. B.Y. and T.A. performed and designed drug combination analyses. T.O. performed pathogen screening analyses. N.S., L.S., P.E.K., D.A.L., T.P.L., H.N., J.S., R.S., Y.H.K., W.S.K., S.-S.C., W.C.C., K.O., and F.I. contributed and prepared biological samples and clinical data. F.I. participated in the study design and supervision of the project. S.M. conceived and designed the study, directed and supervised the research. O.D., M.K., and S.M. wrote the manuscript with contributions from other authors.

## Additional information

**Competing interests:** Labcyte, Inc. and FIMM/University of Helsinki have a collaboration agreement on the utilization of Labcyte's acoustic dispensing technologies. S.M. has received honoraria and research funding from Novartis, Pfizer and Bristol-Myers Squibb (not related to this study). F.I. has received honoraria from Novartis, Asahi Kasei Pharma, Baxalta, Kyowa Hakko-Kirin, Janssen and served as a consultant or in an advisory role for Celgene and Novartis (not related to this study). M.K. has served as consultant for Medix Biochemica (not related to this study). The remaining authors declare no competing interests.

