## [Peer Review File · Nature Communications]

Reviewers' comments:

Reviewer #1 (Remarks to the Author):

Dufva and colleagues have undertaken a study of ANKL, an EBV-associated malignancy of mature NK cells. They use a combination of exome sequencing and screening assays to better understand its biology. I laud the authors on reanalyzing public data to ensure consistent pipelines, and on the first characterization of the genomics of this tumour type and on one of the best methodological reporting of a sequencing dataset this reviewer has encountered recently. Similarly most bioinformatics & statistical analyses are very well-conducted, and I have modest suggestions in that regard. The biology of the study itself is exciting, potentially identifying the origins of ANKL aggressivity and linking nicely cancer genomics to drug-screening in a way that provides key candidates for future mechanistic and translational studies.

1. Technical Sequencing Comments

- * the authors should clarify the average sequencing depth and its standard-deviation
- * the authors should clarify what version of dbSNP was used for filtering
- * please ensure all software version numbers are reported (e.g. bwa-mem for tumour-only samples)
- * it would be useful to see mutations in the tumour types broken out by trinucleotide rather than by base-change (figure 1)
- * it is unclear how sample swap analyses, filtering of OxoG artifacts or lane-wise contamination (e.g. ContEst) were done, those should be specified

2. Tumour-Only Samples

The authors should repeat their procedure for tumour-only samples on the samples with paired tumour/normal to demonstrate the accuracy of the procedure. It is expected that the accuracy will not be perfect, but quantifying this is key to interpreting the results

3. Driver Gene Analysis

While a recurrence of 3/14 patients is clearly quite large for driver genes, the authors should apply formal driver-gene analysis methodology (e.g. ActiveDriver, MutSigCV, OncoDriveFM or equivalents) to control for biases like gene size, replication timing and transcription-coupled repair.

4. Raw Data

Given that a major component of the value of this manuscript is in the sequencing resource it contains, it is key that both the raw reads (e.g. FASTQs or BAMs including unaligned reads) and the variant callset described in this paper are deposited with suitable annotation in appropriate public repositories. This is key to the value of this work.

Reviewer #2 (Remarks to the Author):

The paper has two major parts – firstly carrying out exome sequencing to identify the somatic mutational landscape of ANKL, and secondly carrying out an extensive drug screening on both ANKL and NKTCCL cell lines to identify potential synergistic combinations.

In the first part, 14 ANKL samples undergo exome sequencing (only 4 with matched normal tissue). They identify recurrent mutations in JAK/STAT pathway, RAS/MAPK pathway, and epigenetic regulators. These are the typical pathways that are mutated in other similar leukemia/lymphoma types. Thus, not much novelty for this part. There is also very limited bio-informatic analysis, and no data on expression of the mutated genes. A lot is based on re-analysis of previously published data.

The second part of the manuscript is then focused on NKTCL (not clear why the shift from ANKL to NKTCL). Here, the authors perform drug screens on cell lines to determine sensitivity to 459 known drugs. This is a nice part of the paper, and provides an interesting approach to look for synergy of two drugs. The finding that ruxolitinib (JAK inhibitor) and BCL2 inhibitors work synergistically is of high interest, but also other combinations look promising and should be studied in more detail.

specific comments:

- As part of figure 1b they state that ANKL has a high somatic mutation burden, but this is only based on four ANKL samples, each with very different levels of mutations.
- Figure 1c has a table of selected mutations found in ANKL –also including patients that do not have matched normal samples. It is unclear how they selected these particular genes for the table. More perplexing is the classification of FAT1 and FAT4 as epigenetic modulators? These are large protocadherins with ill-defined function although implicated in wnt- and/or hippo signaling and/or cell adhesion. To this end, it is unclear on the significance of these mutations without associated expression data? E.g. If FAT4 is not expressed – would a mutation in this gene have any effect? A cursory investigation of the expression data (e.g. Immgen) suggests that there is no expression of this gene for example.
- The authors focus on the relationship between ANKL and JAK/STAT mutations and now also include NKTCL as part of the study. Note that on page 4 they state they use 9 cell lines of which 2 are ANKL, but somewhat confusingly in supplementary table 5, three are classified as ANKL? Moreover, the authors carry out RNA-seq for mutational screening to confirm the NK cell lines represent NK cell malignancies. The authors then claim the mutational spectrum between NK cell lines is similar to ANKL and NKTCL – although the data supporting this statement is not entirely clear. Their supplementary figure 4 again has some candidate genes listed as being mutated – but these are different from that presented in Figure 1c. Critically, data showing correlating NKTCL, ANKL and cell lines using RNA-seq data vs wild type NK cells and other normal immune cells from publically available data set would be more essential (I.e. PCA plots showing clustering etc).
- The drug testing part switches somewhat confusingly more toward NKTCL lines. Why is this ?
- In Figure 5 they now only use 7 of the 9 cell lines? Is there any particular reason why two were left out, given one absent cell line is an ANKL cell line (I.e. IMC-1).
- As part of this study, the group uses an improved method for determining synergism and suggest that NKTCL benefit from dual JAK and BCL2 inhibition by using the SNK-6 cell line. However, from the data – it would appear that a large delta score is achieved with the aurora kinase inhibitor alisertib – but this is not followed up. Why? It would add strength to the results if more drug combinations are studied in detail.
- Are there ANKL/NKTCL patient derived xenograft samples available to confirm the findings with the cell lines ? This would strengthen the conclusions.

Reviewer #3 (Remarks to the Author):

Key results: description of mutational landscape in the rare diagnostic entity of aggressive NK cell leukemia / lymphoma. The major claim of this paper is that aggressive NK cell leukemia / lymphoma, EBV positive has a complex mutational landscape affecting JAK-STAT, RAS-MAPK pathways and epigenetic modifiers. The authors attempt to derive practical implications and provide evidence that as a result the cell lines of aggressive NK cell leukemia are sensitive to certain targeted therapies. The results do not fully support the notion that aggressive NK cell leukemia / lymphoma has a distinct mutational profile. There is a large variability of mutations between cases and the set of mutations shows large overlap compared to other diagnostic entities, including NK/T cell lymphoma nasal type. The authors should address in the discussion whether any of their findings could be helpful in

differential diagnosis or serve the purpose of diagnostic test.

The authors are encouraged to apply the nomenclature according to WHO classification of the tumors of the hematopoietic system, for example the term "NK/T cell lymphoma" is not specific, presumably the authors use this term referring to "NK/T cell lymphoma, nasal type", however this is not clear from the text and could be understood differently.

There is a potential clinical overlap between the NK/T cell lymphoma, nasal type and aggressive NK cell leukemia / lymphoma, as well as between the latter and other types of EBV positive lymphoproliferative diseases of NK/T cells. This should be discussed in the text with clarification of the diagnostic criteria and the approach to differential diagnosis.

Overall the biggest drawback I see in this paper is scarcity of clinical and pathologic information about the cases. Where the pathology and clinical records reviewed? what were the inclusion / exclusion criteria? the patient / disease characteristics provided in supplementary table are insufficient. For example: some of the cases are described as CD3 positive, is it by flow cytometry or IHC? if flow cytometry, was it cytoplasmic or surface staining? what was the T cell receptor rearrangement status of these cases?

The diagnosis of aggressive NK cell leukemia / lymphoma should be made per WHO classification. One other aspect of the potential relationship between aggressive NK cell leukemia and NK/T cell lymphoma, nasal type is that there are cases which progress from NKTCL, Nasal type to systemic involvement resembling leukemia, there are speculations in the literature that the two could in fact represent the same disease (per analogy with CLL / SLL), this work has the potential to contribute to this question and this potential has not been fully exploited by the authors. Specifically, based on your data, would you favor the two categories (NK/T cell lymphoma, nasal type and aggressive NK cell leukemia/lymphoma) to represent distinct diagnostic entities or are they similar / closely related?

The role of Epstein Barr virus in the disease pathogenesis is well established, yet the authors do not refer to the relationship between the somatic mutations and EBV, is there any indication whether the somatic mutations are acquired subsequently to the EBV infection or vice versa? recently two papers have performed WES on EBV negative cases of aggressive NK cell leukemia, how do they compare to this study group? are there any potential implications for the pathogenesis of this disease?

Originality: it is not the first report of mutational landscape in aggressive Nk cell leukemia / lymphoma (PMID 27631517, 28548121), however these prior and recent reports focused on the EBV negative cases. Moreover it is a rather rare disease and collection of 14 cases does provide significant novel contribution to the literature of this topic.

Data&methodology: the case selection criteria are not sufficiently explained and patient characteristics included in supplementary table are also not satisfactory.

Figures are well put together and easy to understand.

Validity: I do not see any major flaws prohibiting publication.

- ♣ Originality and significance: a similar study using only cell lines (and not tumor samples) has been reported recently (PMID 28505169), nevertheless the significance is high given presentation of both molecular findings and resulting sensitivity to targeted therapies and would be of interest to broad spectrum of readers.

- ♣ Statistics and methodology of molecular studies: while this is not my primary area of expertise the methods appear sound, moreover the authors have prior record of applying similar methodology and their findings were subsequently confirmed by others, adding validity to their approach.

- ♣ Conclusions: this section could be significantly enhanced, please see suggestions above.

- ♣ The abstract should use unambiguous pathologic terminology (WHO classification)

♣ Literature cited is fairly comprehensive and adequate for this topic
In summary I would recommend giving the authors the opportunity to revise and improve the paper.

Reviewers' comments:

Reviewer #1 (Remarks to the Author):

“Dufva and colleagues have undertaken a study of ANKL, an EBV-associated malignancy of mature NK cells. They use a combination of exome sequencing and screening assays to better understand its biology. I laud the authors on reanalyzing public data to ensure consistent pipelines, and on the first characterization of the genomics of this tumour type and on one of the best methodological reporting of a sequencing dataset this reviewer has encountered recently. Similarly most bioinformatics & statistical analyses are very well-conducted, and I have modest suggestions in that regard. The biology of the study itself is exciting, potentially identifying the origins of ANKL aggressivity and linking nicely cancer genomics to drug-screening in a way that provides key candidates for future mechanistic and translational studies.”

#We thank the reviewer for the positive feedback on our study.

“1. Technical Sequencing Comments

* the authors should clarify the average sequencing depth and its standard-deviation”

#We have now added Supplementary Table 2 describing technical details of the sequencing experiments, including average sequencing depth and its standard deviation. We have evaluated the mean coverage by examining reads mapped to all exons of protein coding genes (Ensembl v82) with 150 bp padding on each side of the exons for consistent results between ANKL and NKTCL samples, as the target enrichment kits differed between these samples. We also report the mean target coverage reported by our sequencing laboratory for the target enrichment kits used in our samples. It should be noted that although the mean coverage of some paired tumor-normal samples was low due to small amount of material available, sorted NK tumor cell fractions were used in exome sequencing to increase the sensitivity of mutation detection. The coverage calculation is also described in Methods (page 13, lines xx).

“* the authors should clarify what version of dbSNP was used for filtering”

#We have added this information in Supplementary Table 2 describing technical aspects of the performed sequencing experiments, in the “Tool list” tab.

“* please ensure all software version numbers are reported (e.g. bwa-mem for tumour-only samples)”

#We have added this information in Supplementary Table 2 in the “Tool list” tab.

“* it would be useful to see mutations in the tumour types broken out by trinucleotide rather than by base-change (figure 1)”

#We thank the reviewer for the insightful suggestion which we believe led to more in-depth understanding of the genetic basis of these tumor types. We have now analyzed the trinucleotide contexts of identified mutations in ANKL in comparison to other related cancers (NKTCL samples from Jiang et al. and T-LGLL, CLPD-NK and T-PLL samples from our previous studies) using the deconstructSigs tool. We found that ANKL samples largely lacked signature 3, associated with failure of DNA double-strand break repair by homologous recombination. Signature 3 was found in most NKTCL and T-LGLL samples, which we found relatively similar to each other in their spectrum of mutational signatures. In contrast, T-PLL samples showed a distinct signature profile dominated by signature 1. These findings also lend credibility to the applied methodology in extracting mutational signatures, as differences in ANKL vs. other tumors do not seem to be associated with the laboratory where sample preparation and sequencing were performed as demonstrated by the similarity of signatures of NKTCL and T-LGL that are from different

laboratories, and the ability to detect differences between samples from the same laboratory (ANKL, CLPD-NK, T-LGLL and T-PLL). We found that ANKL samples tended to cluster together largely owing to the lack of signature 3 when the analysis was performed with also tumor-only samples included to increase the number of samples. Although we acknowledge that there may be caveats in performing such analysis without filtering out germline variants, we believe that absence of a signature is unlikely to be an artefact of the tumor-only samples, as such artefacts would rather be expected to result in the detection of an additional signature because of unfiltered germline variants. Thus, our analysis suggests that ANKL samples differ from NKTCL and other studied T/NK neoplasms regarding underlying mutational processes.

“* it is unclear how sample swap analyses, filtering of OxoG artifacts or lane-wise contamination (e.g. ContEst) were done, those should be specified”

#We have now specified this information in the Methods part (page 14, lines 444-449 for cross-sample contamination and lines 449-451 for OxoG artifacts.). These analysis steps are also outlined in the step-by-step overview of variant calling in Supplementary Table 2.

“2. Tumour-Only Samples

The authors should repeat their procedure for tumour-only samples on the samples with paired tumour/normal to demonstrate the accuracy of the procedure. It is expected that the accuracy will not be perfect, but quantifying this is key to interpreting the results”

#To enable feasible comparisons between the tumor-only and paired samples and to further increase the consistency of the used variant calling methods across data sets, we have now performed variant calls for all samples using the pipeline previously used for the tumor-only samples (while still utilizing the control samples), relying on the GATK best practices. Using these methods, we identified nearly all of the previously highlighted mutations, suggesting that the findings are robust and not dependent on the specific variant calling method used. Accuracy of the tumor-only variant calls based on comparison with the matched tumor-normal variant calls across all WES samples analysed in the study is given in Supplementary Table 2, revealing an average sensitivity of 0.72 and positive predictive value of 0.36 across all tumor-normal pairs included in this study. This is also reported in the Results section (page 5, lines 138-142).

Using the new variant calling method, we were able to identify previously undetected mutations, including one *KMT2D* and one *DDX3X* mutation in ANKL and 2 *STAT3*, one *JAK2*, one *DDX3X* mutation in NKTCL. These findings have now been added to Figures 1d, 2d, Supplementary Figure 4a and Supplementary Tables 5. Supplementary Tables 3 and 4 have been updated to show the results of the new pipeline.

“3. Driver Gene Analysis

While a recurrence of 3/14 patients is clearly quite large for driver genes, the authors should apply formal driver-gene analysis methodology (e.g. ActiveDriver, MutSigCV, OncoDriveFM or equivalents) to control for biases like gene size, replication timing and transcription-coupled repair.”

#We have now applied the MutSigCV and Oncodrive-fm driver gene analyses and provide this information in Supplementary Table 5 and for selected genes in Figure 1d. We note that due to the limited size of the cohort, there is insufficient power for MutSigCV to detect driver genes that would be statistically significant after correction for multiple testing. Therefore the uncorrected P values are reported for MutSigCV in Figure 1d. Furthermore, the application of these tools on tumor-only samples has caveats. As the analyses have to be run on variant lists that are not filtered for their functionality (i.e. synonymous variants must be included), the tumor-only samples contain many false positives (actual germline variants) when the stringent filtering criteria that we demonstrated to efficiently reduce false positives are not utilized. Thus, we have reported the results of these tools for the genes highlighted in Figure 1d based on biological relevance, but do not find it feasible to identify driver genes solely based on these analyses because of the risk of false positives and

small sample size. The driver analyses are commented on in the Results (pages 5-6, lines 168-170).

“4. Raw Data

Given that a major component of the value of this manuscript is in the sequencing resource it contains, it is key that both the raw reads (e.g. FASTQs or BAMs including unaligned reads) and the variant callset described in this paper are deposited with suitable annotation in appropriate public repositories. This is key to the value of this work.”

#All sequencing data from the cell lines is being deposited to the GEO database. Unfortunately, the patient sequence data (WES data and RNA-seq data from normal NK cells) is only available from the corresponding author upon request, owing to regulations pertaining to the authors ethics permit (sequence data not allowed to be deposited in the public repositories). The variant callsets are provided in Supplementary Tables 3, 4 and 9.

Reviewer #2 (Remarks to the Author):

“The paper has two major parts – firstly carrying out exome sequencing to identify the somatic mutational landscape of ANKL, and secondly carrying out an extensive drug screening on both ANKL and NKTCL cell lines to identify potential synergistic combinations.

In the first part, 14 ANKL samples undergo exome sequencing (only 4 with matched normal tissue). They identify recurrent mutations in JAK/STAT pathway, RAS/MAPK pathway, and epigenetic regulators. These are the typical pathways that are mutated in other similar leukemia/lymphoma types. Thus, not much novelty for this part. There is also very limited bio-informatic analysis, and no data on expression of the mutated genes. A lot is based on re-analysis of previously published data.”

#Analysis of somatic mutation landscape in ANKL has not been performed before, and therefore we consider that our manuscript provides significant amount of novel information. We acknowledge the relative similarity of mutational spectrum of ANKL to related leukemia/lymphoma types and consider this finding to provide important novel understanding of the relationships of these diseases at the molecular level. In addition, we have now significantly enhanced our bioinformatics analysis (as detailed in other responses) and have added data on the expression of the mutated genes as suggested. We also wish to highlight that through reanalysis of previously published data we were able to make new discoveries such as mutations and gains in JAK-STAT genes as well as comparisons of mutational processes between cancer types, demonstrating that we did not merely replicate the previous findings but extracted substantial new information.

“The second part of the manuscript is then focused on NKTCL (not clear why the shift from ANKL to NKTCL). Here, the authors perform drug screens on cell lines to determine sensitivity to 459 known drugs. This is a nice part of the paper, and provides an interesting approach to look for synergy of two drugs. The finding that ruxolitinib (JAK inhibitor) and BCL2 inhibitors work synergistically is of high interest, but also other combinations look promising and should be studied in more detail.”

#In the part focusing on drug sensitivity data in the revised manuscript, we have now focused more equally on both ANKL and NKTCL as well as both the JAK/AURK inhibitor and JAK/BCL-2 inhibitor combinations, which both were highlighted by the delta synergy score analysis as correctly pointed out by the reviewer. We have performed additional combination experiments with cell lines that were missing in the previous version, including one ANKL cell line (IMC-1), as suggested by the reviewer. This data is presented in Figure 5, Supplementary Figures 7 and 8, and on pages 8-9, lines 273-284.

“specific comments:

- As part of figure 1b they state that ANKL has a high somatic mutation burden, but this is only based on four ANKL samples, each with very different levels of mutations.”

#We agree with the reviewer that the limited number of ANKL samples precludes definitive conclusions about the mutation burden relative to other cancers, and according to the updated variant calls the difference of ANKL mutation load to other cancers is not statistically significant. We have now modified this statement accordingly (page 4, lines 123-126).

“- Figure 1c has a table of selected mutations found in ANKL –also including patients that do not have matched normal samples. It is unclear how they selected these particular genes for the table. More perplexing is the classification of FAT1 and FAT4 as epigenetic modulators? These are large protocadherins with ill-defined function although implicated in wnt- and/or hippo signaling and/or cell adhesion. To this end, it is unclear on the significance of these mutations without associated expression data? E.g. If FAT4 is not expressed – would a mutation in this gene have any effect? A cursory investigation of the expression data (e.g. Immgen) suggests that there is no expression of this gene for example.”

#The genes presented in the table are selected based on both recurrence and likely biological relevance to tumorigenesis based on the literature. This is now also stated in the table legend. Relevant references to the literature on the mutated genes is presented in the Results section where the mutations are described. We consider these criteria to identify genes that are most likely to be pathogenic in these patients. Highlighting genes based on high mutation frequency would not distinguish likely drivers such as *STAT3* and *DDX3X* from genes that are highly likely irrelevant, including genes encoding large proteins easily accumulating mutations such as mucins. This is demonstrated by the top recurrently mutated genes in our ANKL cohort below:

Mutations	Gene
5	MUC4
4	MUC12
4	DDX3X
3	STAT3
3	NEB
3	MUC5AC
3	MUC22

On the other hand, due to the limited size of the cohort, algorithms assessing the statistical significance of genes being drivers such as MutSigCV and Oncodrive-fm miss several highly likely pathogenic genes such as *NRAS* and *KRAS*, although the codon 12 and 13 mutations identified in these genes are highly likely relevant to tumorigenesis in the ANKL cases harboring these mutations. Thus, we chose to highlight genes in Figure 1d based on the criteria above and report full lists of identified mutations in Supplementary Table 2 and the results of driver analyses in Supplementary Table 5.

We have now also refined the filtering criteria of the tumor-only mutation calls to require all variants to be either reported recurrent in COSMIC or likely to damage the protein structure, in addition to being rare in the healthy population. Using these criteria, one *FAT1* mutation and the *FAT4* mutation are filtered out. Moreover, we agree with the reviewer that utilizing gene expression data is useful to better understand the potential functional significance of the mutations in NK cells. To this end, we have added expression estimates of the highlighted mutations to Figure 1d and to Supplementary Figure 4 as reads per kilobase per million (RPKM) of reads in RNA-seq libraries prepared from the normal NK cells (n=4) and NK cell lines (n=9) profiled in this study. This revealed rather negligible expression of *PTPN21* and low levels of *FAT1*, which have now been removed from Figure 1d based on these criteria but remain in Supplementary Table 3. However,

we note that filtering genes solely based on their expression level may have caveats, as is the case with *PTPRK* that is relatively lowly expressed in our data but has been demonstrated to have a tumor suppressor function in NK cells¹. Finally, we have added three genes identified as mutated by the updated variant calling pipeline to Figure 1d, *PTPN4*, *KMT2D (MLL2)* and *MSH6*. We acknowledge that the mutations presented in Figure 1d are not selected based on unbiased statistical methods for reasons detailed above, and the reader is referred to Supplementary Table 3 for a complete listing of mutations identified by our methods.

“- The authors focus on the relationship between ANKL and JAK/STAT mutations and now also include NKTCL as part of the study. Note that on page 4 they state they use 9 cell lines of which 2 are ANKL, but somewhat confusingly in supplementary table 5, three are classified as ANKL?”

#We apologize for this confusion. We have corrected this information in the text. The correct number of ANKL cell lines is 3.

“Moreover, the author carry out RNA-seq for mutational screening to confirm the NK cell lines represent NK cell malignancies. The authors then claim the mutational spectrum between NK cell lines is similar to ANKL and NKTCL – although the data supporting this statement is not entirely clear. Their supplementary figure 4 again has some candidate genes listed as being mutated – but these are different from that presented in Figure 1c. Critically, data showing correlating NKTCL, ANKL and cell lines using RNA-seq data vs wild type NK cells and other normal immune cells from publically available data set would be more essential (i.e. PCA plots showing clustering etc).”

#To further investigate how the NK cell lines represent primary NK-cell malignancies, we have now improved the comparison of the cell line mutations to primary ANKL and NKTCL mutations, and carried out RNA-seq-based transcriptomic comparisons. For comparison of the mutational spectrum between NK cell lines and primary ANKL and NKTCL samples, we have now added a side-by-side comparison to Supplementary Figure 4a, which contains alterations that are shared between these sample groups and considered functionally relevant based on the literature. Several mutations that are relatively rare in cancer cell lines in general but found commonly in primary ANKL and NKTCL, such as *STAT3*, *STAT5B*, *DDX3X*, *FAS* and *PRDM1*, are found in the cell lines, supporting the statement that the cell lines harbor characteristics of the primary tumor types focused on in this study.

We agree with the reviewer that transcriptomic data are more likely to comprehensively characterize the cell states and phenotypes of the NK cell lines in comparison to primary samples, as opposed to mutations found in a set of genes. To this end, we compared transcriptomes of the cell lines profiled in this study to primary normal NK cells, primary NKTCL samples (n=17), primary normal CD8+ (n=5) and CD4+ (n=3) T cells and primary T-LGLL samples (n=17). These comparisons are presented in Supplementary Figure 4b and c. In principal component analysis of transcriptomes (Supplementary Fig. 4b), the first principal component (PC) largely reflected the difference between T and NK cells. The NK cell lines clustered together with normal NK cells and primary NKTCL along this first PC, indicating that the cell lines represent the NK lineage. Moreover, the cell lines clustered closer to primary NKTCL than the normal NK cells, suggesting that the cell lines transcriptomically resemble primary NK-cell malignancies. Similar results were obtained using unsupervised hierarchical clustering (Supplementary Fig. 4c). These analyses are also included in the Results (page 7, lines 204-215).

“- The drug testing part switches somewhat confusingly more toward NKTCL lines. Why is this ?”

#We have now provided more information on the ANKL cell lines and added results to the text and also to Figure 5 and Supplementary Figures 7 and 8.

“- In Figure 5 they now only use 7 of the 9 cell lines? Is there any particular reason why two were left out, given one absent cell line is an ANKL cell line (i.e. IMC-1).”

#In the original submission, two cell lines (IMC-1 and KAI3) were not included in the drug combination experiments due to logistical reasons: we obtained these cells only after testing all other cell lines in the combination experiments which require custom drug plate preparation as opposed to the single-agent drug screening which is performed routinely. We agree with the reviewer that these additional cell lines add value to the data, particularly as IMC-1 is one of the rare ANKL cell lines. We have now performed combination experiments on the two missing cell lines. In addition, we have now replicated the full 7x7 dose-response matrix combination experiments for all cell lines to ensure consistent and reliable measures of drug interactions. This data is now presented in Figure 5 and Supplementary Figures 7 and 8.

“- As part of this study, the group uses an improved method for determining synergism and suggest that NKTCL benefit from dual JAK and BCL2 inhibition by using the SNK-6 cell line. However, from the data – it would appear that a large delta score is achieved with the aurora kinase inhibitor alisertib – but this is not followed up. Why? It would add strength to the results if more drug combinations are studied in detail. “

#Initially we chose to emphasize the JAK/BCL2 inhibitor combination based on our results that these compounds are most specific towards NK cells compared to other cell types and would thus likely target essential survival mechanisms in NK-cell malignancies. However, the aurora kinase inhibitor alisertib indeed showed synergy comparable to that of venetoclax in combination with ruxolitinib based on the delta score. We have now added results of the JAK/AURK inhibitor combination alongside the JAK/BCL2 inhibitor results in representative cell lines of both ANKL (KHYG-1) and NKTCL (SNK-6). Moreover, drug combination sensitivity and synergy landscape heatmaps of all combinations in all cell lines have been added to Supplementary Figures 7 and 8. These comparisons demonstrate that some cell lines are more sensitive to the JAK/BCL2 combination, whereas others to the JAK/AURK combination. However, our in-house drug sensitivity profiling data suggest that AURK/PLK inhibitors are rather broadly active in highly proliferating samples such as most cancer cell lines, which may indicate lack of specificity but rather a broad antiproliferative effect. This may underlie the rather modest single-agent and combination efficacy in the primary bone marrow lymphoproliferation patient sample compared to the JAK/BCL2 combination. These aspects are now discussed in the Results (page 9, lines 273-284) and Discussion (page 11, lines 340-368).

“- Are there ANKL/NKTCL patient derived xenograft samples available to confirm the findings with the cell lines ? This would strengthen the conclusions.”

#Despite several efforts, we have not been able to obtain viable primary ANKL/NKTCL samples for drug sensitivity profiling. We are also unaware of publications describing patient-derived xenografts of ANKL/NKTCL samples. While we agree that patient-derived xenograft models would allow more definite conclusions to be made from the drug sensitivity profiling experiments, we feel that establishing a xenograft model of ANKL/NKTCL would be outside the scope of this study.

Reviewer #3 (Remarks to the Author):

“Key results: description of mutational landscape in the rare diagnostic entity of aggressive NK cell leukemia / lymphoma. The major claim of this paper is that aggressive NK cell leukemia / lymphoma, EBV positive has a complex mutational landscape affecting JAK-STAT, RAS-MAPK pathways and epigenetic modifiers. The authors attempt to derive practical implications and provide evidence that as a result the cell lines of aggressive NK cell leukemia are sensitive to certain targeted therapies.

The results do not fully support the notion that aggressive Nk cell leukemia / lymphoma has a distinct mutational profile. There is a large variability of mutations between cases and the set of mutations shows large overlap compared to other diagnostic entities, including NK/T cell

lymphoma nasal type. The authors should address in the discussion whether any of their findings could be helpful in differential diagnosis or serve the purpose of diagnostic test.”

We agree with the reviewer that mutations in same genes as found mutated in ANKL in this study are found in related diagnostic entities. However, although many of the mutated genes are similar in ANKL and NKTCL, the new data added in the manuscript (trinucleotide contexts of identified mutations) suggest that ANKL seem to differ from most NKTCL cases in their mutational profile. We have now addressed in the discussion similarities and differences between ANKL and NKTCL as well as other related cancers, and the implications of this to differential diagnosis based on molecular alterations (pages 10-11, lines 312-338).

“The authors are encouraged to apply the nomenclature according to WHO classification of the tumors of the hematopoietic system, for example the term "NK/T cell lymphoma" is not specific, presumably the authors use this term referring to "NK/T cell lymphoma, nasal type", however this is not clear from the text and could be understood differently.”

#We acknowledge the ambiguity of the term “NK/T-cell lymphoma”, which has now been corrected to “extranodal NK/T-cell lymphoma, nasal type” according to the 2016 revision of the WHO classification of lymphoid neoplasms. As stated in the revised abstract, the abbreviation NKTCL is used in the text to refer to this diagnosis.

“There is a potential clinical overlap between the NK/T cell lymphoma, nasal type and aggressive NK cell leukemia / lymphoma, as well as between the latter and other types of EBV positive lymphoproliferative diseases of NK/T cells. This should be discussed in the text with clarification of the diagnostic criteria and the approach to differential diagnosis.

#We have now addressed in the discussion similarities and differences between ANKL and NKTCL (pages 10-11, lines 312-338). Especially based on the trinucleotide contexts of identified mutations it is likely that many ANKL and NKTCL cases arise through different mutational processes, although the resulting alterations required for NK cell transformation may be largely similar in both diseases.

“Overall the biggest drawback I see in this paper is scarcity of clinical and pathologic information about the cases. Where the pathology and clinical records reviewed ? what were the inclusion / exclusion criteria ?”

#We agree with the reviewer that the clinical information presented in the original manuscript was scarce. We have now amended this and provide more detailed information in Supplementary Table 1. Our case collection was started from patients diagnosed at Shinshu University, Japan and then extended to collaborating institutes Shimane University, Japan, Fukuoka University Hospital, Japan, Chi-Mei Medical Center, Taiwan and Samsung Medical Center, South Korea. All available samples with verified ANKL diagnosis and sufficient quality and quantity for next-generation sequencing were identified. ANKL diagnosis was made according to the WHO classification, with a surface immunophenotype characteristic of NK cells (CD2+CD3-CD56+), systemic involvement including bone marrow and/or peripheral blood and an aggressive clinical course as minimum requirements for diagnosis. During the review process, all clinicians and pathologists were contacted again and the data was validated. These criteria are now included in the Methods (page 12, lines 379-389).

“the patient / disease characteristics provided in supplementary table are insufficient. For example: some of the cases are described as CD3 positive, is it by flow cytometry or IHC ? if flow cytometry, was it cytoplasmic or surface staining ? what was the T cell receptor rearrangement status of these cases ?”

#More detailed clinical information has now been added to Supplementary Table 1. For the cases having surface CD3 expression marked as 2% to 29%, analysis was performed by flow cytometry and the values indicate percentages out of lymphocytes, not total leukemic cells, as now clarified in the legend accompanying the table. Therefore, the CD3 positive fraction likely represents non-neoplastic T cells. The T-cell receptor rearrangement status of these cases was found negative, further confirming the NK-cell phenotype of the neoplastic cells. We agree that the presentation of these data was insufficient and this has now been corrected in Supplementary Table 1.

“The diagnosis of aggressive NK cell leukemia / lymphoma should be made per WHO classification.”

#As explained above, ANKL diagnosis was made according to the WHO classification, with a surface immunophenotype characteristic of NK cells (CD2+CD3-CD56+), systemic involvement including bone marrow and/or peripheral blood and an aggressive clinical course as minimum requirements for diagnosis. This is now described in the Methods (page 12, lines 379-389).

“One other aspect of the potential relationship between aggressive NK cell leukemia and NK/T cell lymphoma, nasal type is that there are cases which progress from NKTCL, Nasal type to systemic involvement resembling leukemia, there are speculations in the literature that the two could in fact represent the same disease (per analogy with CLL / SLL), this work has the potential to contribute to this question and this potential has not been fully exploited by the authors. Specifically, based on your data, would you favor the two categories (NK/T cell lymphoma, nasal type and aggressive NK cell leukemia/lymphoma) to represent distinct diagnostic entities or are they similar / closely related ?

#We have now addressed in the discussion similarities and differences between ANKL and NKTCL (pages 10-11, lines 312-338). Especially based on the trinucleotide contexts of identified mutations it is likely that many ANKL and NKTCL cases arise through different mutational processes. However, overall the mutation profile is shared between ANKL and NKTCL and also with some T-cell malignancies, likely reflecting pathways which are important in these cell types.

“The role of Epstein Barr virus in the disease pathogenesis is well established, yet the authors do not refer to the relationship between the somatic mutations and EBV, is there any indication whether the somatic mutations are acquired subsequently to the EBV infection or vice versa ? “

We are not aware of methods to explicitly address whether EBV infection of the cells preceded the acquisition of somatic mutations or vice versa. Particularly, as the clinical records confirmed that all ANKL and NKTCL patients included in the study were positive for EBV, comparison of the somatic mutational profiles relative to EBV status is not feasible within these diseases but only in comparison with other, EBV negative related cancer types. To get insight into potential connections between somatic mutations and EBV, we have now analyzed the mutational profiles, particularly trinucleotide context signatures in relation to the EBV positivity confirmed by WES-based EBV load (sequencing reads mapping to the EBV genome). We did not discover signatures that would be more prevalent in EBV positive tumors (ANKL, NKTCL) compared to EBV negative (T-LGLL, CLPD-NK, T-PLL) as demonstrated by lack of clustering of mutational signatures by EBV status in Supplementary Fig. 4. This information is now presented in Figure 1d and Supplementary Figure 2 and also added in the Results (page 4, lines 128-129) and Discussion parts (page 10, lines 328-331). Therefore, EBV at least does not appear to induce a distinct type of mutations in NK cells. However, based on this data we are not able to answer which occurred first (EBV infection or mutations).

“recently two papers have performed WES on EBV negative cases of aggressive NK cell leukemia, how do they compare to this study group ? are there any potential implications for the pathogenesis of this disease ?”

#We assume that the reviewer refers to Gao et al.² and Nicolae et al.³ papers, although both Gao et al. and Nicolae et al. used rather limited targeted sequencing panels instead of WES. Nevertheless, Nicolae et al. found STAT3 mutations in 2 EBV-negative ANKL cases, whereas Gao et al. found no JAK-STAT pathway-associated gene mutations in 3 EBV-negative cases. Furthermore, Gao et al. also discovered one *DDX3X*, two *TET2* mutations and an *IDH2* hotspot mutation in an EBV-negative ANKL case. Thus, several mutations found in our cohort, such as *STAT3*, *DDX3X* and *TET2* mutations can be found both in EBV-positive and negative cases, suggesting that these genes and EBV can independently contribute to NK-cell transformation. We thank the reviewer for highlighting these studies and have now added the above points to the Discussion part (page 9, lines 295-296 and page 10, lines 328-331).

“Originality: it is not the first report of mutational landscape in aggressive Nk cell leukemia / lymphoma (PMID 27631517, 28548121) , however these prior and recent reports focused on the EBV negative cases. Moreover it is a rather rare disease and collection of 14 cases does provide significant novel contribution to the literature of this topic.”

#As stated above, in the referred papers only very few patients were studied (n=5 in total) and only limited targeted gene panels were used. In addition, those cases were EBV negative as more commonly ANKL is EBV positive. Thus, our results add significant novelty.

“Data&methodology: the case selection criteria are not sufficiently explained and patient characteristics included in supplementary table are also not satisfactory.”

#We have added the case selection criteria in the Methods (page 12, lines 379-389) and extended patient clinical information (Supplementary Table 1).

Figures are well put together and easy to understand.

#We thank for the comment and are happy that the figures are easy to understand.

“Validity: I do not see any major flaws prohibiting publication.

□ Originality and significance: a similar study using only cell lines (and not tumor samples) has been reported recently (PMID 28505169), nevertheless the significance is high given presentation of both molecular findings and resulting sensitivity to targeted therapies and would be of interest to broad spectrum of readers.”

#We thank the reviewer for appreciating the combined genetic and functional analyses performed in this study. We wish to further point out that the study by Mondejar et al. referred above included only two NK cell lines, which were screened for mutations in a 16-gene panel and protein expression of 23 immunomarkers. We consider the characterization molecular alterations of 14 primary ANKL cases by WES and transcriptome-wide mutation identification in 9 NK cell lines with functional interrogation of drug responses of over 400 drugs to provide significantly more comprehensive and unbiased understanding of the molecular pathogenesis and targeted therapy candidates in NK-cell malignancies.

“Statistics and methodology of molecular studies: while this is not my primary area of expertise the methods appear sound, moreover the authors have prior record of applying similar methodology and their findings were subsequently confirmed by others, adding validity to their approach.”

#We thank for the comment and the bioinformatic pipelines are now even further enhanced as detailed above.

“Conclusions: this section could be significantly enhanced , please see suggestions above.”

#We agree that conclusions and interpretation of the data was lacking in the previous letter format of the manuscript, mainly due to space limitations. As detailed above, we have now added a discussion section that addresses the conclusions especially related to the relationship between ANKL and NKTCL and role of EBV in ANKL pathogenesis.

“The abstract should use unambiguous pathologic terminology (WHO classification)”

#This has now been corrected as discussed above.

“Literature cited is fairly comprehensive and adequate for this topic
In summary I would recommend giving the authors the opportunity to revise and improve the paper.”

#We have also added the publications describing previous sequencing studies in EBV-negative ANKL highlighted by the reviewer to the reference list.

1. Chen, Y.-W. *et al.* Receptor-type tyrosine-protein phosphatase κ directly targets STAT3 activation for tumor suppression in nasal NK/T-cell lymphoma. *Blood* **125**, 1589–1600 (2015).
2. Gao, J. *et al.* EBV-negative aggressive NK-cell leukemia/lymphoma: a clinical and pathological study from a single institution. *Modern Pathology* **30**, 1100–1115 (2017).
3. Nicolae, A. *et al.* EBV-negative Aggressive NK-cell Leukemia/Lymphoma: Clinical, Pathologic, and Genetic Features. *Am. J. Surg. Pathol.* (2016). doi:10.1097/PAS.0000000000000735

REVIEWERS' COMMENTS:

Reviewer #1 (Remarks to the Author):

The authors have addressed all my concerns.

Reviewer #2 (Remarks to the Author):

no new comments

Reviewer #3 (Remarks to the Author):

Thank you for the responses and for modifying the paper accordingly. I have no further questions or comments. In my opinion the article is worthy publishing in its current form.